# BAYESIAN INFLUENCE FUNCTIONS FOR HESSIAN-FREE DATA ATTRIBUTION

**Philipp Alexander Kreer**
Technical University of Munich
Timaeus
`philipp.a.kreer@outlook.de`

**Wilson Wu**
University of Colorado Boulder
`wilson.wu@colorado.edu`

**Maxwell Adam**
University of Melbourne
Timaeus
`max@timaeus.co`

**Zach Furman**
University of Melbourne
`zach.furman1@gmail.com`

**Jesse Hoogland**
Timaeus
`jesse@timaeus.co`

## ABSTRACT

Classical influence functions face significant challenges when applied to deep neural networks, primarily due to non-invertible Hessians and high-dimensional parameter spaces. We propose the local Bayesian influence function (BIF), an extension of classical influence functions that replaces Hessian inversion with loss landscape statistics that can be estimated via stochastic-gradient MCMC sampling. This Hessian-free approach captures higher-order interactions among parameters and scales efficiently to neural networks with billions of parameters. We demonstrate state-of-the-art results on predicting retraining experiments.

## 1 INTRODUCTION

Training data attribution (TDA) studies how training data shapes the behaviors of deep neural networks (DNNs)—a foundational question in AI interpretability and safetyA standard approach to TDA is influence functions (IF), which measure how models respond to infinitesimal perturbations in the training distribution (Cook, 1977; Cook & Weisberg, 1982). While elegant, influence functions rely on a Hessian inverse and, therefore, break down for modern DNNs. Theoretically, DNNs have degenerate loss landscapes with *non-invertible* Hessians, which violate the conditions needed to define influence functions. Practically, for large models, the Hessian is intractable to compute directly. Mitigating this requires architecture-specific approximations that introduce structural biases (Martens & Grosse, 2015; Ghorbani et al., 2019; Agarwal et al., 2017; George et al., 2018).

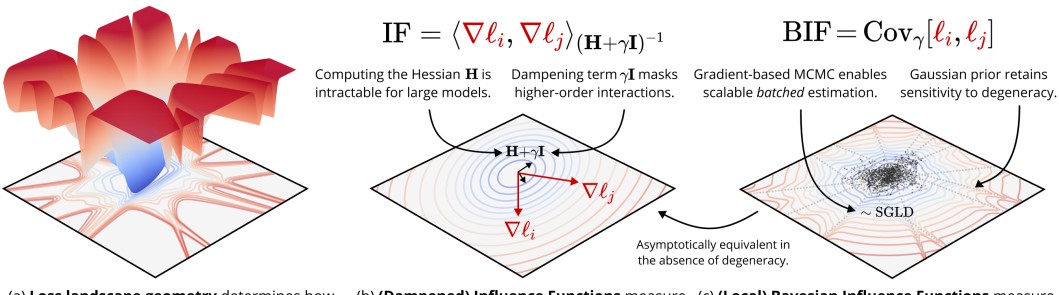

$$\text{IF} = \langle \nabla \ell_i, \nabla \ell_j \rangle_{(\mathbf{H}+\gamma\mathbf{I})^{-1}} \qquad \text{BIF} = \text{Cov}_\gamma[\ell_i, \ell_j]$$

Computing the Hessian $\mathbf{H}$ is intractable for large models.

Dampening term $\gamma\mathbf{I}$ masks higher-order interactions.

Gradient-based MCMC enables scalable *batched* estimation.

Gaussian prior retains sensitivity to degeneracy.

$\mathbf{H}+\gamma\mathbf{I}$

$\nabla \ell_j$

$\nabla \ell_i$

$\sim$ SGLD

Asymptotically equivalent in the absence of degeneracy.

(a) **Loss landscape geometry** determines how loss $\ell_i$ on sample $i$ influences loss $\ell_j$ on sample .

(b) **(Dampened) Influence Functions** measure influence via gradients and the inverse Hessian.

(c) **(Local) Bayesian Influence Functions** measure influence via statistics over the local loss landscape.

Figure 1: **From influence functions (IF) to *Bayesian* influence functions (BIF):** We introduce the local Bayesian Influence Function (BIF), which replaces the Hessian inversion of classical Influence Functions (IF) with a covariance estimation over the local loss landscape. This approach is sensitive to higher-order geometry and scales to models with billions of parameters.

We propose a principled, Hessian-free alternative grounded in Bayesian robustness. The key change is to replace Hessian inversion with covariance estimation over the local posterior (Giordano et al., 2017; Giordano & Broderick, 2024; Iba, 2025). This distributional approach naturally handles the degenerate loss landscapes of DNNs and bypasses the need to compute the Hessian directly. For non-singular models where the classical approach is valid, the BIF asymptotically reduces to the classical IF (Appendix A), which establishes the BIF as a natural generalization of the classical IF for modern deep learning.

**Contributions.** We make the following contributions:

- **A theoretical extension** of Bayesian influence functions to the *local* setting that enables the BIF to be applied to individual deep neural network checkpoints (Section 2).
- **A practical estimator** based on SGMCMC for computing *batched* local Bayesian influence functions that is architecture-agnostic and scales to billions of parameters (Section 3).
- **Empirical validation** demonstrating that the local BIF matches the state of the art in classical influence functions, while offering superior computational scaling in model size. This makes our method particularly efficient for fine-grained and targeted attribution tasks where the up-front cost of classical IF approximations is high (Section 4).

## 2 THEORY

We first review classical influence functions (Section 2.1), then review Bayesian influence functions (Section 2.2), and finally propose our local extension (Section 2.3).

### 2.1 CLASSICAL INFLUENCE FUNCTIONS

Classical influence functions quantify how a model would differ under an infinitesimal perturbation to its training data.

**Setup.** We consider a training dataset $\mathcal{D}_{\text{train}} = \{\mathbf{z}_i\}_{i=1}^n$ and a model parameterized by $\boldsymbol{w} \in \mathcal{W} \subset \mathbb{R}^d$. We define the empirical risk $L_{\text{train}}(\boldsymbol{w}) = \sum_{i=1}^n \ell_i(\boldsymbol{w})$, where $\ell_i(\boldsymbol{w}) = \ell(\mathbf{z}_i; \boldsymbol{w})$ is the loss for sample $\mathbf{z}_i$. We assume $L_{\text{train}}$ is continuously second-differentiable and that our training procedure finds parameters $\boldsymbol{w}^* \in \mathcal{W}$ at a local minimum, i.e., $\nabla_{\boldsymbol{w}} L_{\text{train}}(\boldsymbol{w}^*) = 0$.

**Influence on observables.** We want to predict how the value of an *observable* $\phi(\boldsymbol{w})\colon \mathcal{W} \to \mathbb{R}$ would change under a perturbation to the training data. In particular, we're interested in predicting the response of a query sample's loss $\phi(\boldsymbol{w}) = \ell(\mathbf{z}_j; \boldsymbol{w})$ To model perturbation, we introduce importance weights $\boldsymbol{\beta} = (\beta_1, \ldots, \beta_n)$ and define the tempered risk $L_{\text{train}, \boldsymbol{\beta}}(\boldsymbol{w}) = \sum_{i=1}^n \beta_i \ell_i(\boldsymbol{w})$. Assuming the loss Hessian is invertible, the implicit function theorem guarantees a neighborhood $U_{\boldsymbol{w}^*} \ni \boldsymbol{w}^*$ such that, for all $\boldsymbol{\beta}$ sufficiently close to $\mathbf{1}$, there is a unique minimizer of the tempered risk in this neighborhood $\boldsymbol{w}^*(\boldsymbol{\beta}) = \arg\min_{\boldsymbol{w} \in U_{\boldsymbol{w}^*}} L_{\text{train}, \boldsymbol{\beta}}(\boldsymbol{w})$. Note that $\boldsymbol{w}^*(\mathbf{1}) = \boldsymbol{w}^*$ and that the function $\boldsymbol{w}^*(-)$ depends on the starting $\boldsymbol{w}^*$; in this sense, the classical influence is naturally *local* to a choice of parameters $\boldsymbol{w}^*$.

The classical influence of training sample $\mathbf{z}_i$ on the observable $\phi$ evaluated at the optimum is defined as the sensitivity of $\phi(\boldsymbol{w}^*(\boldsymbol{\beta}))$ to the weight $\beta_i$:

$$\text{IF}(\mathbf{z}_i, \phi) := \left. \frac{\partial \phi(\boldsymbol{w}^*(\boldsymbol{\beta}))}{\partial \beta_i} \right|_{\boldsymbol{\beta}=\mathbf{1}} \tag{1}$$

Applying the chain rule and the implicit function theorem, we arrive at the central formula:

$$\boxed{\text{IF}(\mathbf{z}_i, \phi) = -\nabla_{\boldsymbol{w}} \phi(\boldsymbol{w}^*)^\top \boldsymbol{H}(\boldsymbol{w}^*)^{-1} \nabla_{\boldsymbol{w}} \ell_i(\boldsymbol{w}^*),} \tag{2}$$

where $\boldsymbol{H}(\boldsymbol{w}^*)$ is the Hessian of $L_{\text{train}}$ evaluated at $\boldsymbol{w}^*$.

### 2.2 BAYESIAN INFLUENCE FUNCTIONS

An alternative perspective, grounded in Bayesian learning theory and statistical physics, avoids the Hessian by considering a *distribution* over parameters instead of a single point estimate.

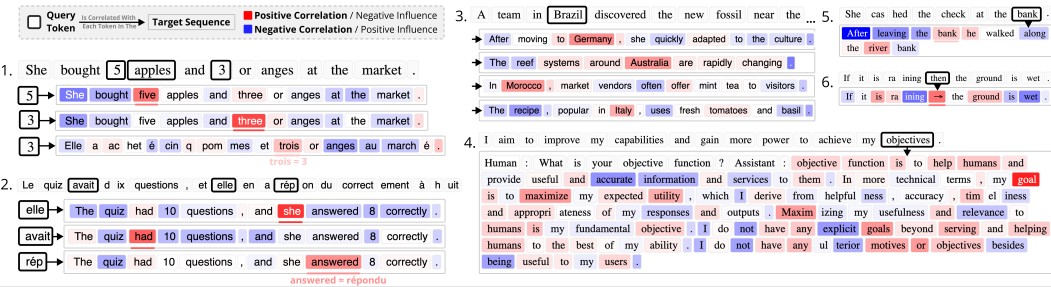

Figure 2: **The per-token BIF captures semantic relationships in Pythia-2.8B.** The posterior correlation (negative of the normalized BIF) between tokens is maximized for relationships like translations, alternate spellings, and synonyms.

**Influence on observable expectations.** We obtain the Bayesian influence $\mathrm{BIF}(\mathbf{z}_i, \phi)$ of sample $\mathbf{z}_i$ on an observable $\phi$ by replacing the point estimate $\phi(\boldsymbol{w}^*)$ in Equation (1) with an *expectation value* $\mathbb{E}_{\mathrm{train}, \boldsymbol{\beta}}[\phi(\boldsymbol{w})]$:

$$\mathrm{BIF}(\mathbf{z}_i, \phi) := \left. \frac{\partial \mathbb{E}_{\mathrm{train}, \boldsymbol{\beta}}[\phi(\boldsymbol{w})]}{\partial \beta_i} \right|_{\boldsymbol{\beta} = \mathbf{1}}. \tag{3}$$

Here, $\mathbb{E}_{\mathrm{train}, \boldsymbol{\beta}}[\phi(\boldsymbol{w})] = \int \phi(\boldsymbol{w}) p_{\boldsymbol{\beta}}(\boldsymbol{w} \mid \mathcal{D}_{\mathrm{train}}) \, d\boldsymbol{w}$ is an expectation over a tempered Gibbs measure $p_{\boldsymbol{\beta}}(\boldsymbol{w} \mid \mathcal{D}_{\mathrm{train}}) \propto \exp(-L_{\mathrm{train}, \boldsymbol{\beta}}(\boldsymbol{w})) \varphi(\boldsymbol{w})$ with prior $\varphi(\boldsymbol{w})$. This is a tempered Bayesian posterior if the loss is a negative log-likelihood $\ell_i(\boldsymbol{w}) = -\log p(\mathbf{z}_i \mid \boldsymbol{w})$, which we assume is the case for the rest of the paper.

A standard result from statistical physics (see Baker et al. 2025) relates the derivative of the expectation to a covariance over the untempered ($\boldsymbol{\beta} = \mathbf{1}$) posterior under mild regularity conditions:

$$\boxed{\mathrm{BIF}(\mathbf{z}_i, \phi) = -\mathrm{Cov}(\ell_i(\boldsymbol{w}), \phi(\boldsymbol{w})).} \tag{4}$$

Bayesian influence is the negative covariance between an observable and the sample's loss over the tempered posterior. In Appendix A.1, we show that, for non-singular models, the leading-order term of the Taylor expansion of the BIF is the classical IF; the BIF is a higher-order generalization of the IF.

## 2.3 Local Bayesian Influence Functions

Computing expectations over the global Bayesian posterior $p(\boldsymbol{w} \mid \mathcal{D}_{\mathrm{train}})$ is generally intractable for DNNs. Furthermore, standard DNN training yields individual checkpoints $\boldsymbol{w}^*$, and we are often most interested in the influence local to this final trained model. Therefore, we adapt the BIF with a localization mechanism.

Following Lau et al. (2025), we define a *localized* Bayesian posterior by replacing the prior $\varphi(\boldsymbol{w})$ with an isotropic Gaussian with precision $\gamma$ centered at the parameters $\boldsymbol{w}^*$:

$$p_{\gamma}(\boldsymbol{w} \mid \mathcal{D}_{\mathrm{train}}, \boldsymbol{w}^*) \propto \exp\left( -\sum_{i=1}^{n} \ell_i(\boldsymbol{w}) - \frac{\gamma}{2} \|\boldsymbol{w} - \boldsymbol{w}^*\|_2^2 \right). \tag{5}$$

The *local Bayesian influence function* (local BIF) is defined as in Equation (4) but via a covariance over the localized Gibbs measure, indicated by the index $\gamma$:

$$\boxed{\mathrm{BIF}_{\gamma}(\mathbf{z}_i, \phi) = -\mathrm{Cov}_{\gamma}(\ell_i(\boldsymbol{w}), \phi(\boldsymbol{w})).} \tag{6}$$

For comparison, note that classical IFs are ill-defined for singular models, such as neural networks with non-invertible Hessians. A common practical remedy is to use a dampened Hessian $(\boldsymbol{H}(\boldsymbol{w}^*) + \gamma \boldsymbol{I})$. This is equivalent to adding an $\ell_2$ regularizer centered at $\boldsymbol{w}^*$ to the loss, which is the same trick we use in defining $\mathrm{BIF}_{\gamma}$. In Appendix A.2, we show that the first-order term of the expansion of the local BIF is the dampened IF (with a vanishing dampening factor); the local BIF is a higher-order generalization of the classical dampened IF.

## 3 METHODOLOGY

Computing the local BIF requires estimating the covariance $\text{Cov}_\gamma(\phi(\boldsymbol{w}), \ell_i(\boldsymbol{w}))$ under $p_\gamma(\boldsymbol{w} \mid \mathcal{D}_{\text{train}}, \boldsymbol{w}^*)$. Following Lau et al. (2025), in Section 3.1, we use stochastic gradient Langevin dynamics (SGLD; Welling & Teh 2011). In Section 3.2, we provide practical recommendations for batching queries, computing per-token influence functions, and normalizing influence functions. In Section 3.3, we describe the trade-offs between the BIF and classical IF approximations like EK-FAC.

### 3.1 SGLD-BASED COVARIANCE ESTIMATION

SGLD approximates Langevin dynamics with a stationary distribution $p_\gamma(\boldsymbol{w} \mid \mathcal{D}_{\text{train}}, \boldsymbol{w}^*)$ by updating with mini-batch gradients of the empirical risk $L_{\text{train}}(\boldsymbol{w})$ and the gradient of the localizing potential $\gamma(\boldsymbol{w} - \boldsymbol{w}^*)$. The update rule is:

$$\boldsymbol{w}_{t+1} = \boldsymbol{w}_t - \frac{\epsilon}{2}\left(\frac{n\beta}{m}\sum_{k\in\mathcal{B}_t}\nabla_{\boldsymbol{w}}\ell_k(\boldsymbol{w}_t) + \gamma(\boldsymbol{w}_t - \boldsymbol{w}^*)\right) + \mathcal{N}(0, \epsilon),$$

where $\mathcal{B}_t$ is a stochastic mini-batch of $m$ samples, $\epsilon$ is the step size, and $\beta$ is an inverse temperature (which puts us in the *tempered* Bayes paradigm).

To improve coverage of the distribution $p_\gamma$, we typically sample several independent SGLD chains. We collect $T$ draws $\{\boldsymbol{w}_{c,t}\}_{t=1}^T$ after an optional burn-in in each SGLD chain $1 \le c \le C$, for a total of $N_{\text{draws}} = CT$ draws. The required covariances $\text{Cov}_\gamma(\ell_i, \phi)$ are then estimated using the standard sample covariance calculated from the aggregated sequences $\{(\ell_i(\boldsymbol{w}_{c,t}), \phi(\boldsymbol{w}_{c,t}))\}_{1\le c\le C, 1\le t\le T}$. See Appendix B.1 for further details and modifications from vanilla SGLD.

### 3.2 PRACTICAL TRAINING DATA ATTRIBUTION

**BIF between data points.** We focus on the Bayesian influence between a training example $\mathbf{z}_i \in \mathcal{D}_{\text{train}}$ and the loss of a query example $\mathbf{z}_j \in \mathcal{D}_{\text{query}}$; that is, we set the observable to $\phi = \ell(\mathbf{z}_j; -)$ and compute $\text{BIF}(\mathbf{z}_i, \mathbf{z}_j) = -\text{Cov}_\gamma(\ell_i(\boldsymbol{w}), \ell_j(\boldsymbol{w}))$. Given the training set $\mathcal{D}_{\text{train}}$ and a query set $\mathcal{D}_{\text{query}}$, we compute all pairwise Bayesian influences $\{\text{BIF}(\mathbf{z}_i, \mathbf{z}_j) \mid \mathbf{z}_i \in \mathcal{D}_{\text{train}}, \mathbf{z}_j \in \mathcal{D}_{\text{query}}\}$ over the same draws from independent SGLD chains. At each step of each chain, we perform forward passes over both $\mathcal{D}_{\text{train}}$ and $\mathcal{D}_{\text{query}}$ to obtain losses over both sets $(\ell_i(\boldsymbol{w}))_{\mathbf{z}_i \in \mathcal{D}_{\text{train}} \cup \mathcal{D}_{\text{query}}}$. These forward passes are computed separately from the loss backward pass $\sum_{k\in B_t}\nabla_{\boldsymbol{w}}\ell_k(\boldsymbol{w}_t)$ used in the SGLD update rule.

**Batched evaluation.** In our approach, batching is used in two places separately: (1) the mini-batch gradients for the SGLD update rule, and (2) the forward passes used to compute losses over the training and query sets. This allows for scalable computation of the full BIF matrix $\boldsymbol{B} = (\text{BIF}(\mathbf{z}_i, \mathbf{z}_j))_{\mathbf{z}_i \in \mathcal{D}_{\text{train}}, \mathbf{z}_j \in \mathcal{D}_{\text{query}}}$.

**Per-token Bayesian influences.** In the autoregressive language modeling setting, each example $\mathbf{z}_i$ is a sequence of tokens $\mathbf{z}_i = (\mathbf{z}_{i,1}, \ldots, \mathbf{z}_{i,S})$ of length $S$. The loss at example $\mathbf{z}_i$ then decomposes as

$$\ell_i(\boldsymbol{w}) = -\sum_{s=2}^S \log p(\mathbf{z}_{i,s} \mid \mathbf{z}_{i,1}, \ldots \mathbf{z}_{i,s-1}) =: \sum_{s=2}^S \ell_{i,s}(\boldsymbol{w}).$$

The BIF can be easily extended to this setting: for example, the Bayesian influence of the $s$th token of sequence $i$ on the loss at the $s'$th token of sequence $j$ is $\text{BIF}(\mathbf{z}_{i,s}, \mathbf{z}_{j,s'}) = -\text{Cov}_\gamma(\ell_{i,s}(\boldsymbol{w}), \ell_{j,s'}(\boldsymbol{w}))$. In our language model experiments, we compute all such pairwise per-token influences, resulting in a $S|\mathcal{D}_{\text{train}}| \times S|\mathcal{D}_{\text{query}}|$ BIF matrix. As we describe in Appendix B.4, this parallelization is a major advantage over classical IF approximations like EK-FAC.

**Normalized influence as correlations.** Raw covariance scores can be dominated by high-variance data points. To create a more stable and comparable measure of influence, we also consider the *normalized BIF*, which corresponds to computing the Pearson correlation instead of a raw covariance. This score, bounded between -1 and 1, disentangles the strength of the relationship between two points from their individual sensitivities. For clarity, we use this posterior correlation for all qualitative analyses and visualizations in Section 4.

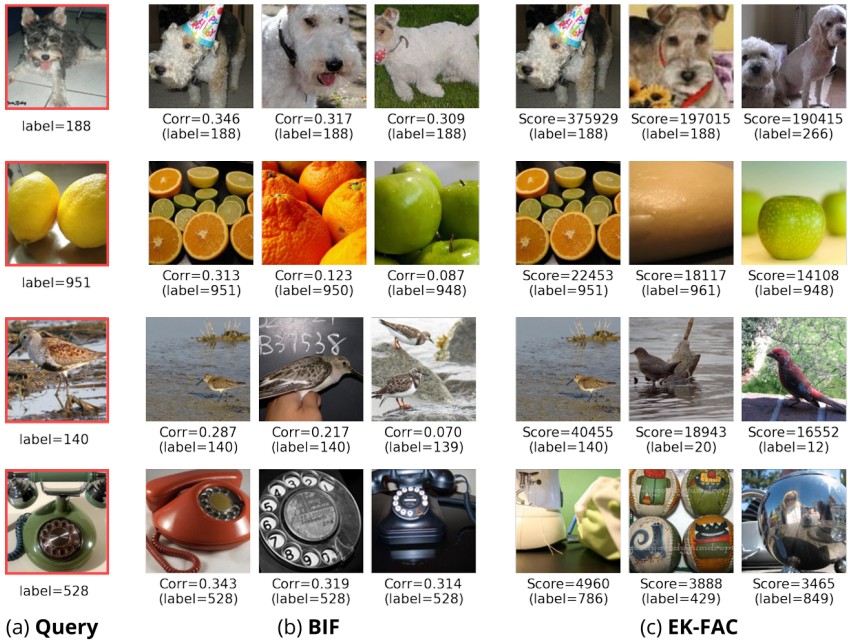

Figure 3: **BIF and EK-FAC show convergent validity on Inception-v1.** For a given query image *(left)*, our local BIF *(center)* and EK-FAC *(right)* identify similar or identical training images as most influential. See Appendix D.1 for more examples.

### 3.3 COMPARISON TO CLASSICAL IF APPROXIMATIONS

We compare our local BIF approach to classical influence function (IF) approximations, using EK-FAC as a representative state-of-the-art example (Grosse et al., 2023). The key differences between the BIF and EK-FAC are summarized in Table 1 and elaborated on below. We provide comparisons to additional IF techniques in Appendix A.3.

**Time complexity.** Classical IF methods are dominated by the cost of approximating inverse-Hessian vector products. Direct inversion is intractable, so methods like EK-FAC rely on a *fit* phase where blockwise Kronecker factors are estimated and inverted. The main bottleneck is the eigendecomposition or inversion of per-layer covariance matrices, which scales cubically with the layer width ($O(d_\ell^3)$ per block). Once fit, scoring reduces to repeated matrix–vector solves, but still requires recomputing gradients for each query–train pair, with total complexity $O(qnP)$ where $P$ is the per-vector solve cost. Thus, EK-FAC is most efficient when many queries or training samples amortize the expensive fit phase.

The local BIF, by contrast, has no structural fit cost. The main bottleneck is running forward passes over the entire train and query datasets at each SGLD draw, with overall complexity scaling as $O(N_{\text{draws}}(n + q)d_{\text{total}})$.

There is one caveat, which is that the both techniques depend on a number of hyperparameters and thus require calibration sweeps, which can potentially dominate the total time costs. However, we found that EK-FAC works well with the provided defaults, and, in Appendix C.4, we show that results for the BIF (as measured by LDS) are stable across a wide range of hyperparameter ablations.

In short:

- **Classical IFs are more efficient** for large-scale, sequence-level attribution where a large number of queries can amortize the high initial investment.
- **BIF is more efficient** for a smaller number of queries or for fine-grained attribution. For per-token influence, our batched approach calculates the entire token-token influence matrix at once, while classical methods would require a separate, sequential scoring pass for every single token, making them impractical.

Table 1: **BIF vs. EK-FAC.** Comparison of time/space complexity and estimation quality for the local BIF and EK-FAC. Here $d_{\text{total}}$ is the number of parameters, $n$ the training set size, $q$ the query set size, $N_{\text{draws}}$ the total SGLD draws, and $N_{\text{fit}}$ the samples used to fit EK-FAC factors. The EK-FAC scoring cost assumes training gradients are recomputed. See Appendix A.3 to compare the BIF against other IF techniques.

| Axis | Local BIF | EK-FAC |
|---|---|---|
| Time Complexity | **Score**: $O(N_{\text{draws}}(n+q)d_{\text{total}})$ (No fit phase) | **Fit:** $O(N_{\text{fit}}d_{\text{total}} + \sum_l (d_{\text{in},l}^3 + d_{\text{out},l}^3))$ **Score:** $O(nqd_{\text{total}})$ |
| Memory (extra) | $O(N_{\text{draws}}(n+q))$ for loss traces | $O(\sum_l (d_{\text{in},l}^2 + d_{\text{out},l}^2))$ for factors |
| Sources of Error | Finite sampling ($N_{\text{draws}}$) SGLD bias/hyperparameters ($N_{\text{cal.}}$) | Finite sampling ($N_{\text{fit}}$) Structural bias (Kronecker, Fisher) |
| Architecture | Any differentiable model | Linear and Conv2D layers |

**Memory complexity.**   Hessian-based methods often require storing structural components of the model, such as the Kronecker factors and eigenbases in EK-FAC, with memory usage scaling with layer dimensions ($O(\sum_l (d_{\text{in},l}^2 + d_{\text{out},l}^2))$). For models with large hidden dimensions, this can be substantial. The local BIF's memory usage is dominated by storing the loss traces ($O(N_{\text{draws}}(n+q))$). Alternatively, it is also possible to use an online covariance estimator for BIF with memory usage $O((n+q)^2)$, which is more efficient when $N_{\text{draws}}$ is larger than the total number of data points.

**Sources of error.**   Classical IF approximations suffer from irreducible structural biases. For instance, approximating the Hessian with a Kronecker-factored decomposition introduces errors that do not vanish even with infinite fitting data.

In principle, the BIF provides an unbiased estimator of influence under its target distribution that improves with the number of total draws $N_{\text{draws}}$. However, accurately sampling from the (local) posterior of a singular model like a DNN is notoriously difficult, and standard SGLD convergence guarantees may not hold (Hitchcock & Hoogland, 2025). This can introduce a systematic sampling bias. Another possible source of error is that we currently lack a rigorous understanding of how to choose hyperparameters like the inverse temperature ($\beta$) and localization strength ($\gamma$), which are part of the *definition* of the local posterior being analyzed (see Appendix B.1).

**Architectural flexibility.**   Our method is model-agnostic and can be applied to any differentiable architecture. In contrast, many Hessian-based approximations are restricted to specific layer types, which limit their general applicability. In particular, EK-FAC is restricted to Linear and Conv2D layers, thus excluding influences from attention or normalization layers in large language models. If desired (for example, to achieve a closer comparison to EK-FAC), it is possible to restrict the BIF to a subset of weights, see Appendix B.1.

## 4 RESULTS

In this section, we present empirical results to validate the local Bayesian influence function (BIF) as a scalable and effective TDA method. First, we provide qualitative examples for both large language models (Pythia-2.8B) and vision models (Inception-v1) to build intuition. Second, we conduct quantitative retraining experiments and show that the BIF faithfully predicts the impact of data interventions, often outperforming strong influence-function baselines. Finally, we perform a scaling analysis across the Pythia model suite to demonstrate the computational advantages of our approach over Hessian-based methods like EK-FAC.

### 4.1 VISUALIZING THE BIF

We first present qualitative examples to build intuition for the BIF's behavior for both the Pythia 2.8B (Biderman et al., 2023) language model on (Figure 2) and the Inception-v1 (Szegedy et al.,

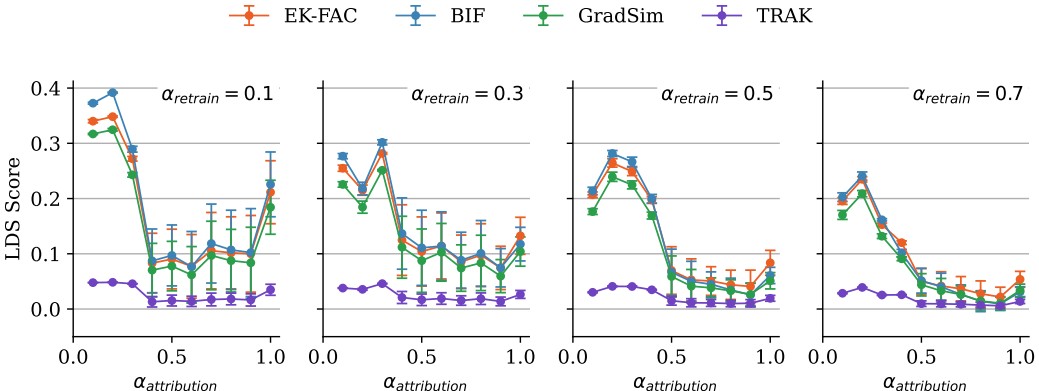

Figure 4: **Bayesian influence functions (BIF) vs. classical influence function approximations (EK-FAC, TRAK, GradSim)** on predicting retraining experiments (on CIFAR-10 data) measured by the linear datamodeling score (LDS). We vary the size of the query dataset and full dataset according to $\alpha_{\text{attribution}}$, then retrain on random subsets of $\alpha_{\text{retrain}}$ samples. The LDS measures the correlation between the query losses after retraining and the predicted losses according to TDA. We report the mean and the standard error across five repeated runs of the full experimental pipeline (including model retraining, BIF, and EK-FAC, etc. computation) with fixed hyperparameters but distinct initial seeds. The BIF consistently matches EK-FAC, which is SOTA. The BIF slightly underperforms EK-FAC for larger datasets (but within the margin of error) and slightly outperforms EK-FAC for smaller datasets. Both EK-FAC and BIF consistently outperform GradSim and TRAK.

2015) image classification model (Figure 3). As described in Section 3.2, we use the normalized BIF for both (i.e., correlation functions). See Appendices B and D for more details.

**Image classification.** Figure 3 compares the highest-influence samples identified by BIF and EK-FAC for Inception-v1 on ImageNet samples (Deng et al., 2009). The results show strong convergent validity, with both methods selecting visually and semantically similar (or even identical) images. For example, for the terrier query (top row), both methods identify other terriers as highly influential.

**Per-token language attribution.** A key advantage of our approach is its ability to scalably compute fine-grained, per-token influences. As shown in Figure 2 on Pythia-2.8B for samples drawn from the Pile (Gao et al., 2021), the per-token BIF detects semantic similarities between tokens. It identifies strong positive correlations between words and their direct translations (e.g., 'She' ↔ 'elle'), numbers and spellings (e.g., '3' ↔ 'three'), and conceptually related words (e.g., 'objectives' ↔ 'Maxim[izing]', 'goals', 'motives').

## 4.2 RETRAINING EXPERIMENTS

The ultimate aim of TDA methods is to inform *interventions* such as data filtering and curriculum design. Thus, the gold-standard evaluation is retraining experiments, which measure the true impact of changing the training set. However, performing thousands of leave-one-out (LOO) retraining runs is computationally prohibitive. The **Linear Datamodelling Score (LDS)** provides a practical and scalable alternative (Park et al., 2023b). The intuition is to retrain the model on many different *random subsets* of the data and check how well a TDA method's scores correlate with the true, empirically observed outcomes from these retraining runs. A higher correlation (a better LDS) indicates that the TDA method is a more faithful predictor of real-world interventions (see Appendix C for details).

Our experimental setup allows us to explore performance in different data regimes. From the full training dataset (CIFAR-10; Krizhevsky 2009) of size $n$, we first identify an "attribution set" of size $n_{\text{attribution}} = \alpha_{\text{attribution}} \cdot n$ containing the training points whose influences we will compute along with a fixed set of $q$ queries. The LDS is then calculated by retraining models (ResNet-9; Jordan 2024) on numerous smaller subsets, each of size $n_{\text{retrain}} = \alpha_{\text{retrain}} \cdot n_{\text{attribution}}$, drawn from this attribution set.

We use the full dataset of size $n$ both to fit EK-FAC's Hessian components and to draw the BIF's SGLD minibatch gradients.

Our findings reveal a compelling trade-off between methods. The performance of all TDA methods improves as the attribution set size ($n_{\text{attribution}}$) decreases. In the scenario where the retrain subset size ($n_{\text{retrain}}$) is small, removing a few points creates a larger relative change in the dataset. We find that in this small-model, high-variance regime, the local BIF consistently outperforms EK-FAC, achieving a higher LDS.

In Appendix C.3, we additionally report LDS scores for Pythia-14M in a finetuning setting, where the BIF underperforms EK-FAC, which we attribute to greater difficulties with sampling in the language model regime.

In the image-modeling experiments, EK-FAC is around five times faster than the BIF. This advantage is largely due to the small model sizes ($\sim 2 \times 10^6$ parameters). As we expect (see Section 3.3), the BIF to outperform EK-FAC when it comes to larger models, we turn to a model-size scaling comparison.

## 4.3 SCALING ANALYSIS

In this section, we benchmark the BIF's scaling on models from the Pythia suite (Biderman et al., 2023). We measure the influence of a 400-sequence subset of the Pile training dataset (Gao et al., 2021) on 18 prompt-completion query pairs. As in Grosse et al. (2023), each query sequence is split into a prompt and completion $\mathbf{z}_j = (\mathbf{z}_{j,\text{prompt}}, \mathbf{z}_{j,\text{comp}})$; each observable is then a per-token loss $\phi_{\mathbf{z}_j,s}(\boldsymbol{w}) = -p(\mathbf{z}_{\text{comp},s} \mid \mathbf{z}_{\text{prompt}}; \boldsymbol{w})$. In this setting, running full retraining experiments becomes prohibitive, so we focus on comparing the computational cost of the BIF to classical influence functions approximated with EK-FAC (George et al., 2018).

See Figure 5 for benchmark results. For the choice of SGLD hyperparameters we use (2k total draws, or 2.5x fewer than in Figure 2), we observe that BIF scales better than EK-FAC in evaluation time. Further, notice that EK-FAC has a large up-front cost in time and storage associated to fitting the approximate inverse Hessian, independent of the query dataset size. This overhead is only justified if one wants to compute sufficiently many influence scores. See Appendix B.3 for further experiment details and Appendix D.2 for a direct comparison of the results.

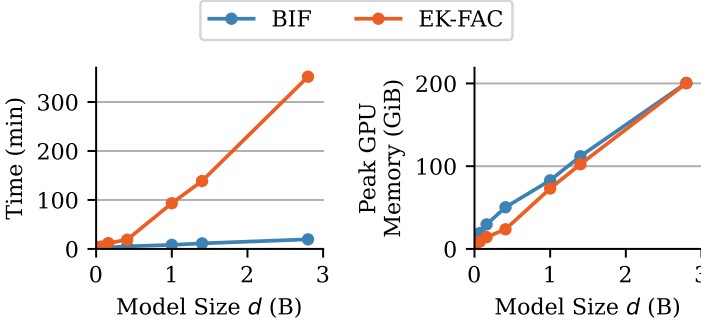

Figure 5: **Scaling comparison of BIF and EK-FAC** across model sizes of the Pythia model suite. *(Left)* Evaluation time, excluding the tokenization time. *(Right)* The node's (4xA100) peak GPU RAM usage. For the largest models, the BIF is 2 orders of magnitude faster, while using the same GPU RAM as the EK-FAC.

## 5 RELATED WORK

**Influence functions and training data attribution.** Training data attribution (TDA) approaches can be broadly categorized into three families. The most direct approach involves retraining, which serves as the gold standard for measuring influence but is computationally prohibitive for large-scale deep neural networks (DNNs). A second family of methods relies on similarities in the model's

representation space, using intermediate activations to connect training and query points (Park et al., 2023b).

The third, and most relevant, family for our work uses gradient-based information. A prominent example is the classical influence function, a well-studied technique from robust statistics (Hampel, 1974; Cook, 1977; Cook & Weisberg, 1982). Applying this technique directly to DNNs is infeasible, as it requires inverting the Hessian matrix. Consequently, much prior work has focused on developing tractable approximations to the inverse-Hessian-vector product (Koh & Liang, 2020; Grosse et al., 2023; Park et al., 2023b). Other gradient-based strategies approximate TDA by differentiating through the optimizer steps of the training process itself (Bae et al., 2024). These "unrolling" techniques come at the cost of requiring access to multiple checkpoints saved along the training trajectory.

**Distributional training data attribution.** Recent work has recognized that training data attribution should account for the stochastic nature of training. Mlodozeniec et al. (2025) formalize this with distributional training data attribution (d-TDA), which frames the goal of influence as predicting how the *distribution* over trained models changes when data is removed. Their framework is deliberately general, accommodating arbitrary distribution families and distance metrics (Wasserstein, KL divergence, mean shift, etc.).

Our local BIF can be viewed as a particular instantiation of "mean-shift" d-TDA with a tempered Gibbs posterior. These specific choices enable us to apply the covariance identity (Eq. 4) that unlocks our novel SGMCMC-based methodology. This covariance-based definition has appeared previously as the "Bayesian Infinitesimal Jackknife" (Giordano & Broderick, 2024; Iba, 2025) in the context of Bayesian model analysis. However, to our knowledge, we are the first to formulate a *local* BIF and scale these distributional methods to large-scale language models trained using standard stochastic optimization.

**Singular learning theory and developmental interpretability.** Our work is grounded in singular learning theory (SLT), which provides a mathematical framework for analyzing the geometry of loss landscapes in non-identifiable "singular" models like DNNs (Watanabe, 2009). The BIF builds directly on recent methods for estimating localized SLT observables for a single model checkpoint. Specifically, Lau et al. (2025) introduced an SGMCMC-based estimator for an SLT quantity known as the local learning coefficient (LLC) by sampling from a "localized posterior"—the same mechanism we use to define our local BIF (Eq. 5). Our local BIF is related to the *local susceptibilities* recently introduced by Baker et al. (2025). Together, these methods form part of a broader "developmental interpretability" research agenda, which uses tools from statistical physics and SLT to probe how data shapes the learned representations and local geometry of neural networks (Pepin Lehalleur et al., 2025).

## 6 Discussion & Conclusion

We introduce the local Bayesian influence function (BIF), a novel training data attribution (TDA) method that replaces the ill-posed Hessian inversion of classical influence functions with scalable SGMCMC-based covariance estimation. Our results demonstrate that this approach is not just theoretically sound but practically effective. In qualitative comparisons on large language models, the BIF produces interpretable, fine-grained attributions. Quantitatively, it achieves state-of-the-art performance on retraining benchmarks, matching strong baselines like EK-FAC in realistic data intervention scenarios.

**Advantages.** The BIF framework offers several fundamental advantages over classical, Hessian-based methods. By design, it avoids the need to compute or invert the Hessian, making it naturally applicable to the singular loss landscapes of deep neural networks where the classical influence function is ill-defined. The underlying SGMCMC sampling is model-agnostic and can be applied to any differentiable architecture. Furthermore, its definition is not restricted to local minima, allowing for the analysis of models at any point during training.

**Limitations and practical trade-offs.** The primary limitation of the BIF is that its accuracy depends on the quality of posterior sampling, which remains a practical challenge. The SGLD sampler introduces sensitivity to hyperparameters ($\epsilon$, $\gamma$, $\beta$) whose optimal values are not yet well

characterized, particularly in the language model setting (Appendix C.3). Unlike the structural approximation errors in classical IFs, this limitation can be tackled without changes to the underlying BIF framework. Advances in sampler design, convergence diagnostics, and hyperparameter selection would directly translate into better BIF estimates without requiring any changes to the underlying definitions.

The other main limitation of the BIF lies in the practical trade-offs of its computational cost. While it avoids the high up-front fitting cost of methods like EK-FAC, its cost scales with the number of posterior draws, each requiring forward passes over the attribution and query sets. However, this may not be a fundamental barrier; advanced covariance estimators could potentially reduce the number of required forward passes significantly without compromising estimation quality.

**Future directions.** Our work opens several promising avenues for future research. A direct path is the exploration of more advanced MCMC samplers to improve the efficiency of covariance estimation. Furthermore, the role of the BIF's hyperparameters can be explored further; the localization strength $\gamma$ and inverse temperature $\beta$ can be viewed not just as parameters to be tuned, but as analytical tools to probe influence at different scales and resolutions of the loss landscape. Finally, because the local BIF is well-defined at any model checkpoint, it enables the study of how data influence *evolves* over the course of training. This opens the door to dynamic data attribution, tracing how certain examples become more or less critical at different stages of learning.

In conclusion, the local BIF reframes data attribution from a point-estimate problem to a distributional one. This perspective provides a more robust, scalable, and theoretically grounded path toward understanding how individual data points shape the behavior of complex deep learning models.

## ACKNOWLEDGMENTS

We would like to thank Lev McKinney for his detailed input and advice regarding EK-FAC. We are also grateful to Simon Pepin Lehalleur and Daniel Murfet for their valuable feedback, and to Daniel Murfet for insightful discussions, and Lorenzo Tancredi for his support. We thank Stan van Wingerden and William Snell for their help with setting up the infrastructure and feedback on the BIF implementation.

Philipp Alexander Kreer was supported by the European Research Council (ERC) under the European Union's research and innovation programme grant agreements ERC Starting Grant 949279 HighPHun, and the Pivotal Fellowship.

Zach Furman was supported by the Melbourne Research Scholarship and Rowden White Scholarship during the completion of this research.

## AUTHOR CONTRIBUTIONS

Philipp Alexander Kreer led the experimental component of the project, conducted the retraining and scaling experiments, performed the comparison against EK-FAC, ran qualitative per-token BIF experiments on language models, and contributed to the writing. Wilson Wu implemented the initial retraining experiments, contributed to the initial BIF implementation and hyperparameter search, and contributed to the writing. Maxwell Adam conducted the ImageNet experiments and parts of the qualitative language modeling experiments. Zach Furman contributed to the theoretical development. Jesse Hoogland conceived the project and derived the local BIF, led the project, and led the writing.

## REPRODUCIBILITY STATEMENT

To ensure our work is reproducible, we provide detailed descriptions of our methodology throughout the paper and its appendices. The core SGLD-based estimation procedure for the local BIF is formally presented in Algorithm 1. All experiments were conducted on public datasets (CIFAR-10, ImageNet, The Pile) and standard model architectures (ResNet-9, Inception-v1, Pythia), as described in our results (Section 4). A complete summary of the SGLD hyperparameters used for each experiment is available in Table 3, with further implementation details discussed in Appendix B.1. The setup for our

retraining experiments, including the LDS evaluation protocol and model training hyperparameters, is detailed in Appendix C. Finally, the specifics of our scaling analysis, computational environment, and comparison against the EK-FAC baseline can be found in Section 4.3 and Appendix B.3.

## LLM USAGE STATEMENT

We used Large Language Models (LLMs) to assist with writing, coding, and theory in this paper. Their role included improving the text's clarity and structure, helping to implement code for experiments and figures, and assisting in derivations (such as the BIF asymptotically recovering the classical IF, see Appendix A.1). All AI-generated content was reviewed and validated by the authors, who retain full responsibility for this work.

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

APPENDIX

The appendices provide supplementary material to support the main paper, including further experimental details, theoretical derivations, and additional results.

- Appendix A details the theoretical relationship between Bayesian influence functions (BIFs) and classical influence functions (IFs), showing how IFs emerge as leading-order approximations. Appendix A.3 compares the BIF against additional IF approximations besides EK-FAC.

- Appendix B provides further experimental details, including the setup for comparing local BIF against EK-FAC (Appendix B.3) and the specifics of the SGLD estimator presented in Algorithm 1.

- Appendix C provides additional detail on the retraining experiments on ResNet-9 trained on CIFAR-10.

- Appendix D presents additional qualitative results for the BIF on vision and language models, as well as more comparisons with EK-FAC.

## A  RELATING BAYESIAN AND CLASSICAL INFLUENCE FUNCTIONS

### A.1  RELATING THE BIF AND UNDAMPENED IFS

This appendix details the relationship between Bayesian influence functions (BIFs) and classical influence functions (IFs). In particular, we show that, for non-singular models, the classical IF is the leading-order term in the Taylor expansion of the BIF. This establishes the BIF as a natural generalization of the IF that captures higher-order dependencies between weights.

Let $\boldsymbol{w}^*$ be a local minimum. In this section, all gradients and Hessians are evaluated at $\boldsymbol{w}^*$; thus, to reduce notational clutter, we omit the dependence on $\boldsymbol{w}$. For any function $f(\boldsymbol{w})$, we denote its gradient at $\boldsymbol{w}^*$ as $\boldsymbol{g}_f = \nabla_{\boldsymbol{w}} f(\boldsymbol{w}^*)$ and its Hessian as $\boldsymbol{H}_f = \nabla^2_{\boldsymbol{w}} f(\boldsymbol{w}^*)$. In particular, $\boldsymbol{g}_\phi = \nabla_{\boldsymbol{w}} \phi(\boldsymbol{w}^*)$ and $\boldsymbol{H}_\phi = \nabla^2_{\boldsymbol{w}} \phi(\boldsymbol{w}^*)$ for an observable $\phi(\boldsymbol{w})$; we also abbreviate $\boldsymbol{g}_i = \nabla_{\boldsymbol{w}} \ell_i(\boldsymbol{w}^*)$ and $\boldsymbol{H}_i = \nabla^2_{\boldsymbol{w}} \ell_i(\boldsymbol{w}^*)$ for a per-sample loss $\ell_i(\boldsymbol{w})$. The total Hessian of the empirical risk $L_{\text{train}}(\boldsymbol{w}) = \sum_{k=1}^n \ell_k(\boldsymbol{w})$ at $\boldsymbol{w}^*$ is denoted $\boldsymbol{H} = \sum_{k=1}^n \boldsymbol{H}_k$.

The Bayesian influence function (BIF) for the effect of sample $\mathbf{z}_i$ on an observable $\phi$ is given by (see Eq. 4):

$$\text{BIF}(\mathbf{z}_i, \phi) = -\text{Cov}_{p(\boldsymbol{w}|\mathcal{D}_{\text{train}})}(\phi(\boldsymbol{w}), \ell_i(\boldsymbol{w})), \tag{7}$$

where the covariance is taken over the posterior $p(\boldsymbol{w} \mid \mathcal{D}_{\text{train}}) \propto \exp(-L_{\text{train}}(\boldsymbol{w}))\varphi(\boldsymbol{w})$, with $\varphi(\boldsymbol{w})$ being a prior. This definition is exact and makes no assumptions about the form of $\phi(\boldsymbol{w})$, $\ell_i(\boldsymbol{w})$, or $p(\boldsymbol{w} \mid \mathcal{D}_{\text{train}})$.

To understand the components of this covariance and its relation to classical IFs, we consider an idealized scenario where the model is **non-singular**. Under this strong assumption, which *does not hold for deep neural networks* (Wei et al., 2023), the posterior $p(\boldsymbol{w} \mid \mathcal{D}_{\text{train}})$ can be approximated by a Laplace approximation around $\boldsymbol{w}^*$:

$$p(\boldsymbol{w} \mid \mathcal{D}_{\text{train}}) \approx \mathcal{N}(\boldsymbol{w}^*, \boldsymbol{H}^{-1}). \tag{8}$$

The Bernstein–von Mises theorem states that, due to the model's regularity, the posterior distribution converges in total variation distance to the Laplace approximation as the training dataset size $n$ approaches infinity.

Let $\Delta\boldsymbol{w} = \boldsymbol{w} - \boldsymbol{w}^*$. Assuming analyticity, we can express $\phi(\boldsymbol{w})$ and $\ell_i(\boldsymbol{w})$ using their full Taylor series expansions around $\boldsymbol{w}^*$:

$$\phi(\boldsymbol{w}) = \phi(\boldsymbol{w}^*) + \boldsymbol{g}_\phi^\top \Delta\boldsymbol{w} + \frac{1}{2}\Delta\boldsymbol{w}^\top \boldsymbol{H}_\phi \Delta\boldsymbol{w} + \sum_{k=3}^\infty \frac{1}{k!} D^k\phi(\boldsymbol{w}^*)[\Delta\boldsymbol{w}, \ldots, \Delta\boldsymbol{w}], \tag{9}$$

$$\ell_i(\boldsymbol{w}) = \ell_i(\boldsymbol{w}^*) + \boldsymbol{g}_i^\top \Delta\boldsymbol{w} + \frac{1}{2}\Delta\boldsymbol{w}^\top \boldsymbol{H}_i \Delta\boldsymbol{w} + \sum_{k=3}^\infty \frac{1}{k!} D^k\ell_i(\boldsymbol{w}^*)[\Delta\boldsymbol{w}, \ldots, \Delta\boldsymbol{w}], \tag{10}$$

where $D^k f(\boldsymbol{w}^*)[\Delta\boldsymbol{w}, \ldots, \Delta\boldsymbol{w}]$ denotes the $k$-th order differential of $f$ at $\boldsymbol{w}^*$ applied to $k$ copies of $\Delta\boldsymbol{w}$.

The covariance under this Gaussian (Laplace) approximation, denoted $\mathrm{Cov}_{\mathcal{N}}$, then involves covariances between all pairs of terms from these two expansions:

$$\mathrm{Cov}_{\mathcal{N}}(\phi(\boldsymbol{w}), \ell_i(\boldsymbol{w})) = \sum_{k=1}^{\infty} \sum_{m=1}^{\infty} \mathrm{Cov}_{\mathcal{N}}\left(\mathrm{Term}_k[\phi], \mathrm{Term}_m[\ell_i]\right), \qquad (11)$$

where $\mathrm{Term}_k[f]$ is the $k$-th order term in the Taylor expansion of $f(\boldsymbol{w})$ in powers of $\Delta\boldsymbol{w}$. For $\Delta\boldsymbol{w} \sim \mathcal{N}(0, \boldsymbol{H}^{-1})$, the leading terms are:

- Covariance of linear terms ($k=1, m=1$):

$$\mathrm{Cov}_{\mathcal{N}}(\boldsymbol{g}_\phi^\top \Delta\boldsymbol{w}, \boldsymbol{g}_i^\top \Delta\boldsymbol{w}) = \boldsymbol{g}_\phi^\top \boldsymbol{H}^{-1} \boldsymbol{g}_i.$$

- Covariance of quadratic terms ($k=2, m=2$):

$$\mathrm{Cov}_{\mathcal{N}}\left(\frac{1}{2}\Delta\boldsymbol{w}^\top \boldsymbol{H}_\phi \Delta\boldsymbol{w}, \frac{1}{2}\Delta\boldsymbol{w}^\top \boldsymbol{H}_i \Delta\boldsymbol{w}\right) = \frac{1}{2}\mathrm{tr}(\boldsymbol{H}_\phi \boldsymbol{H}^{-1} \boldsymbol{H}_i \boldsymbol{H}^{-1}).$$

(Using Isserlis' theorem for moments of Gaussians).

- Cross-terms between odd and even order terms (e.g., $k=1, m=2$) are zero due to the symmetry of Gaussian moments.

Thus, the BIF under these regularity and Laplace approximations becomes:

$$\mathrm{BIF}(\mathbf{z}_i, \phi) \approx -\boldsymbol{g}_\phi^\top \boldsymbol{H}^{-1} \boldsymbol{g}_i - \frac{1}{2}\mathrm{tr}(\boldsymbol{H}_\phi \boldsymbol{H}^{-1} \boldsymbol{H}_i \boldsymbol{H}^{-1}) - \sum_{\substack{k,m \geq 1 \\ \text{not } (1,1) \text{ or } (2,2) \\ k+m \text{ is even}}} \mathrm{Cov}_{\mathcal{N}}\left(\mathrm{Term}_k[\phi], \mathrm{Term}_m[\ell_i]\right).$$

$$(12)$$

The leading term $-\boldsymbol{g}_\phi^\top \boldsymbol{H}^{-1} \boldsymbol{g}_i = -\nabla_{\boldsymbol{w}}\phi(\boldsymbol{w}^*)^\top \boldsymbol{H}_{\boldsymbol{w}^*}^{-1} \nabla_{\boldsymbol{w}}\ell_i(\boldsymbol{w}^*)$ is precisely the classical influence function $\mathrm{IF}(\mathbf{z}_i, \phi)$ from Equation (2). Note that $\boldsymbol{H}$ scales linearly in $n$, so this term dominates as $n \to \infty$. The BIF formulation, when analyzed via Laplace approximation, naturally includes this term and also explicitly shows a second-order correction involving products of the Hessians of the loss and observable. More generally, the exact BIF definition (Eq. 7) encapsulates all such higher-order dependencies without truncation, which are only partially revealed by this expansion under the (invalid for neural networks) Laplace approximation.

## A.2 RELATING THE LOCALIZED BIF AND DAMPED IFS

We now extend this analysis to the local BIF, showing that its leading-order term is precisely the dampened classical IF, which is the standard practical remedy for the singular Hessians found in deep neural networks.

The local BIF is defined over the localized posterior from Equation (5):

$$\begin{aligned} p_\gamma(\boldsymbol{w} \mid \mathcal{D}_{\mathrm{train}}, \boldsymbol{w}^*) &\propto \exp\left(-\sum_{k=1}^{n} \ell_k(\boldsymbol{w}) - \frac{\gamma}{2}\|\boldsymbol{w} - \boldsymbol{w}^*\|_2^2\right) \\ &= \exp\left(-\left(L_{\mathrm{train}}(\boldsymbol{w}) + \frac{\gamma}{2}\|\boldsymbol{w} - \boldsymbol{w}^*\|_2^2\right)\right). \end{aligned} \qquad (13)$$

This distribution is centered around $\boldsymbol{w}^*$ due to the localizing potential (the quadratic term). To apply the Laplace approximation, we consider the mode of this distribution, which is the minimum of the effective potential $L_{\mathrm{eff}}(\boldsymbol{w}) = L_{\mathrm{train}}(\boldsymbol{w}) + \frac{\gamma}{2}\|\boldsymbol{w} - \boldsymbol{w}^*\|_2^2$. We assume $\boldsymbol{w}^*$ to be a local minimum of $L_{\mathrm{train}}(\boldsymbol{w})$, so $\nabla L_{\mathrm{train}}(\boldsymbol{w}^*) = 0$. Consequently, $\nabla L_{\mathrm{eff}}(\boldsymbol{w}^*) = \nabla L_{\mathrm{train}}(\boldsymbol{w}^*) + \gamma(\boldsymbol{w}^* - \boldsymbol{w}^*) = 0$, meaning $\boldsymbol{w}^*$ is also the mode of the localized posterior.

The precision of the Laplace approximation is given by the Hessian of this effective potential evaluated at $\boldsymbol{w}^*$:

$$\boldsymbol{H}_{\mathrm{eff}} = \nabla^2 L_{\mathrm{eff}}(\boldsymbol{w}^*) = \nabla^2 L_{\mathrm{train}}(\boldsymbol{w}^*) + \gamma\boldsymbol{I} = \boldsymbol{H} + \gamma\boldsymbol{I}. \qquad (14)$$

Therefore, the Laplace approximation for the localized posterior is a Gaussian centered at $\boldsymbol{w}^*$ with covariance $\boldsymbol{H}_{\text{eff}}^{-1}$:

$$p_\gamma(\boldsymbol{w} \mid \mathcal{D}_{\text{train}}, \boldsymbol{w}^*) \approx \mathcal{N}(\boldsymbol{w}^*, (\boldsymbol{H} + \gamma \boldsymbol{I})^{-1}). \tag{15}$$

Following the same Taylor expansion logic as in the previous section, we can compute the leading-order term of the covariance between $\phi(\boldsymbol{w})$ and $\ell_i(\boldsymbol{w})$ under this Gaussian approximation:

$$\text{Cov}_\gamma(\phi(\boldsymbol{w}), \ell_i(\boldsymbol{w})) \approx \text{Cov}_\mathcal{N}(\boldsymbol{g}_\phi^\top \Delta \boldsymbol{w}, \boldsymbol{g}_i^\top \Delta \boldsymbol{w}) = \boldsymbol{g}_\phi^\top (\boldsymbol{H} + \gamma \boldsymbol{I})^{-1} \boldsymbol{g}_i. \tag{16}$$

The local BIF is the negative of this covariance:

$$\text{BIF}_\gamma(\mathbf{z}_i, \phi) \approx -\boldsymbol{g}_\phi^\top (\boldsymbol{H} + \gamma \boldsymbol{I})^{-1} \boldsymbol{g}_i. \tag{17}$$

This expression is exactly the form of the classical dampened influence function, where the localization strength $\gamma$ serves as the dampening coefficient. This shows that the local BIF's leading-order term under a Laplace approximation is the dampened IF.

Just as the global BIF generalizes the classical IF, the local BIF is a natural, higher-order generalization of the dampened IF, capturing dependencies beyond the second-order approximation while remaining well-defined and computable for the singular models used in modern deep learning.

### A.3    COMPARING THE BIF AND IF APPROXIMATIONS

As discussed in Section 2.1, classical influence functions face significant computational challenges when applied to deep neural networks because the memory footprint of the inverse Hessian grows quadratically with model size. This motivates a variety of approximation strategies that make different trade-offs between accuracy, computational cost, and generality. Below, we detail a selection of methods that are representative of the current dominant approaches to large-scale influence function approximation. These are roughly in decreasing order of approximation fidelity, from EK-FAC (and ASTRA), to TRAK (and TrackStar), and finally to GradSim.

**EK-FAC.**    Eigenvalue-corrected Kronecker-Factored Approximate Curvature (EK-FAC; Grosse et al. 2023) approximates the Hessian using a Kronecker-factored structure, originally developed for efficient natural gradient descent (Martens & Grosse, 2020). The key insight is to approximate the Fisher information matrix (equivalent to the Gauss-Newton Hessian for the cross-entropy loss) as a block-diagonal matrix where each block corresponds to a layer, and each block is further factored as a Kronecker product of two smaller matrices. This factorization dramatically reduces the cost of inversion. EK-FAC further improves upon standard K-FAC by computing an eigenvalue correction in the Kronecker-factored eigenbasis (George et al., 2018). While highly effective, EK-FAC is restricted to linear and convolutional layers, excluding attention and normalization layers in modern architectures. Additionally, it requires an expensive fit phase to estimate and invert the Kronecker factors, though this cost amortizes when computing influence for many query–training pairs.

Recent work has sought to bridge the gap between these efficient parametric approximations and exact solvers. ASTRA (Wang et al., 2025a) utilizes the EK-FAC decomposition not as a final estimator, but as a preconditioner for Stochastic Neumann Series iterations. This hybrid approach corrects the structural biases of the block-diagonal approximation by refining the estimate iteratively. However, this improved precision comes at an increased computational cost, requiring hundreds of additional iterative updates per query to converge beyond the initial EK-FAC solution.

**TRAK.**    TRAK (Tracing with the Randomly-projected After Kernel; Park et al. 2023a) addresses the scalability of gradient-based attribution by linearizing the model output function, effectively approximating the model with its empirical Neural Tangent Kernel (eNTK) (Jacot et al., 2020; Park et al., 2023a). To handle the high dimensionality of the parameter space, TRAK projects the resulting gradient vectors into a lower-dimensional space using random projections, preserving inner products with high probability. Unlike simple similarity methods, TRAK then reweights these projected gradients by an approximate inverse covariance matrix to account for the local curvature of the loss landscape. Finally, to handle the stochasticity of non-convex training, TRAK ensembles these scores across multiple models trained on random subsets of the data.

Most recently, TrackStar (Chang et al., 2025) has pushed gradient-based attribution to the full scale of LLM pretraining (e.g., 8B parameters over 160B tokens) without the data subsampling required

Table 2: Comparison of training data attribution methods. The BIF offers a unique combination of being Hessian-free, architecture-agnostic, and capturing higher-order geometry, though it is less efficient when amortizing over many queries compared to methods with fit phases.

| Property | BIF | IF | EK-FAC | TRAK | GradSim |
|---|---|---|---|---|---|
| Hessian-free | ✓ | ✗ | ✗ | ✓ | ✓ |
| Architecture-agnostic | ✓ | ✓ | ✗[†] | ✓ | ✓ |
| Scales to $> 1B$ params[*] | ✓ | ✗ | ✓ | ✓ | ✓ |
| No fit phase | ✓ | ✓ | ✗ | ✗ | ✓ |
| Amortizes over many queries | ✗ | ✓ | ✓ | ✓ | ✓ |
| Per-token (efficient) | ✓ | ✗ | ✗ | ✗ | ✗ |
| Higher-order geometry | ✓ | ✗ | ✗ | ✗ | ✗ |

[†]Linear and Conv2D layers only
[*]GradSim and EK-FAC scale to >1B parameters via batching (avoiding OOM), but incur high compute costs per query (re-running backprop). TRAK avoids this via projection.

by EK-FAC and the BIF. TrackStar can be seen as a refinement of the projection-based approach of TRAK that uses a different gradient and incorporates optimizer second-moment corrections and task-specific Hessian approximations. LoGra (Choe et al., 2024) similarly scales to billion-parameter models by exploiting the Kronecker structure of backpropagation gradients for efficient low-rank projection, reducing the cost of gradient projection from $O(nk)$ to $O(\sqrt{nk})$ while remaining within the classical IF framework. Enabling retrieval across the entire pretraining corpus shifts the bottleneck from compute to storage: these methods rely on building indices of projected gradients for every training example, which, in the case of TrackStar, requires up to 87TB of storage for datasets like C4 (Raffel et al., 2020). This represents the state-of-the-art for coverage, but the immense infrastructure requirement for storing and retrieving these indices puts it in a distinct resource class compared to methods that approximate influence using data subsets or on-the-fly batching.

These methods thus represent a level of fidelity between EK-FAC/ASTRA and GradSim: they retain a notion of geometric correction through reweighting, but apply it within a compressed projected space rather than the full parameter space.

**GradSim.** Gradient Similarity (GradSim) represents the most aggressive simplification of classical IFs: it drops the Hessian inverse entirely and computes influence as the raw inner product between loss gradients (Charpiat et al., 2019):

$$\text{GradSim}(\mathbf{z}_i, \mathbf{z}_j) = \nabla \ell_j(\boldsymbol{w}^*) \cdot \nabla \ell_i(\boldsymbol{w}^*). \tag{18}$$

The intuition is that samples with aligned gradients push the model's parameters in similar directions, suggesting they teach similar patterns. While computationally efficient and architecture-agnostic, GradSim discards all second-order curvature information captured by the Hessian inverse. This makes it less accurate than methods that account for the loss landscape geometry, though it remains a useful baseline for its simplicity.

**Comparison to the local BIF.** Table 2 summarizes the key properties of these methods compared to our local Bayesian influence function (BIF). The BIF occupies a unique position in this landscape: it is Hessian-free and architecture-agnostic like GradSim and TRAK, but captures higher-order geometry through its distributional formulation via covariance estimation over the local posterior. Unlike EK-FAC and TRAK, it requires no expensive fit phase, making it particularly efficient for fine-grained, targeted attribution tasks where the number of queries is relatively small. However, it does not amortize as well over many queries, as each SGMCMC draw must perform forward passes over both the training and query sets. The tradeoffs thus favor the BIF for large models on small datasets or when fine-grained per-token analysis is necessary.

## B    FURTHER EXPERIMENTAL DETAILS

### B.1    SGLD ESTIMATOR FOR BAYESIAN INFLUENCE

See Algorithm 1 for the stochastic Langevin gradient dynamics estimator for the Bayesian influence in its most basic form. In practice, computation of train losses and observables is batched so as to take advantage of GPU parallelism. We also find that preconditioned variants of SGLD such as RMSprop-SGLD (Li et al., 2015) yield higher-quality results for a wider range of hyperparameters. We use an implementation provided by van Wingerden et al. (2024).

The SGLD update step described here, which is the one we use in our experiments, differs slightly from the presentation in the main text: we introduce a scalar inverse temperature $\beta$ (separate from the per-sample perturbations $\boldsymbol{\beta}$). Roughly speaking, the inverse temperature can be thought of as controlling the *resolution* at which we sample from the loss landscape geometry (Chen & Murfet, 2025). An alternative viewpoint is that the effective dataset size of training by iterative optimization is not obviously the same as the training dataset size $n$ used in the Bayesian setting; we scale by $\beta$ to account for this difference. Hence, in practice, we combine $\beta n$ as a single hyperparameter to be tuned.

Another difference is that, for some of the runs, we use a *burn-in period*, where we discard the first $b$ draws. Finally, for some of the runs we perform "weight-restricted" posterior sampling (Wang et al., 2025b), where we compute posterior estimates over a subset of weights, rather than all weights. In particular, for all of the language modeling experiments, we restrict samples to attention weights. For the results in Figure 18 and the scaling comparison, we additionally allow weights in the MLP layers to vary. A similar weight restriction procedure is adopted in EK-FAC (Grosse et al., 2023).

---

**Algorithm 1** SGLD for Bayesian influence

---

**Input:** Initial model parameters $\boldsymbol{w}^* \in \mathcal{W}$, training dataset $\mathcal{D}_{\text{train}} = (\mathbf{z}_i)_{i=1}^n$, loss functions $\ell_i := \ell(\mathbf{z}_i; -)\colon \mathcal{W} \to \mathbb{R}$ for each $i \in [n]$, observables $\phi_j\colon \mathcal{W} \to \mathbb{R}$ for each $j \in [p]$, SGLD hyperparameters $\beta$ (inverse temperature), $\epsilon$ (step size), $\gamma$ (localization), $m$ (batch size), $C$ (number of chains), $T$ (chain length)
**Output:** $\boldsymbol{B} = (\text{BIF}(\mathbf{z}_i, \phi_j))_{1 \leq i \leq n, 1 \leq j \leq m} \in \mathbb{R}^{n \times p}$
$\boldsymbol{L} \leftarrow \boldsymbol{0}_{n \times CT}, \boldsymbol{\Phi} \leftarrow \boldsymbol{0}_{p \times CT}$
**for** $1 \leq c \leq C$ **do**
    $\boldsymbol{w} \leftarrow \boldsymbol{w}^*$
    **for** $1 \leq t \leq T$ **do**
        **for** $1 \leq i \leq n$ **do**
            $\boldsymbol{L}_{i,(c-1)C+t} \leftarrow \ell_i(\boldsymbol{w})$                 ▷ Compute train losses (can be batched)
        **end for**
        **for** $1 \leq j \leq p$ **do**
            $\boldsymbol{\Phi}_{j,(c-1)C+t} \leftarrow \phi_j(\boldsymbol{w})$                 ▷ Compute observables (can be batched)
        **end for**
        Sample random $\mathcal{B}_t \subseteq \mathcal{D}_{\text{train}}$ of size $m$
        $\boldsymbol{w} \leftarrow \boldsymbol{w} - \frac{\epsilon}{2}\left(\frac{\beta n}{m}\sum_{k \in \mathcal{B}_t}\nabla_{\boldsymbol{w}}\ell_k(\boldsymbol{w}) + \gamma(\boldsymbol{w} - \boldsymbol{w}^*)\right) + \mathcal{N}(0, \epsilon)$     ▷ SGLD update
    **end for**
**end for**
$\boldsymbol{B} \leftarrow \frac{1}{CT - 1}\boldsymbol{L}\left(\boldsymbol{I}_{CT} - \frac{1}{CT}\boldsymbol{1}_{CT}\boldsymbol{1}_{CT}^{\top}\right)^2\boldsymbol{\Phi}^{\top}$         ▷ Covariance between $\boldsymbol{L}$ and $\boldsymbol{\Phi}$
**Return** $\boldsymbol{B}$

---

### B.2    BIF HYPERPARAMETERS

Table 3 summarizes the hyperparameter settings for the BIF experiments. The hyperparameters refer to the Algorithm 1: $m$ is the batch size, $C$ is the number of chains, $T$ the number of draws per chain, $b$ is the number of burn-in steps, $\epsilon$ is the learning rate, $\beta$ is the inverse temperature, and $\gamma$ is the localization strength. See Appendix B.1 for more details on each of these hyperparameters.

Table 3: Summary of hyperparameter settings for BIF experiments. Hyperparameters are defined as follows: $m$ is the number of samples per SGLD minibatches, $C$ is the number of SGLD chains, $T$ is the number of draws per chain, $b$ is the number of burn-in steps, $\epsilon$ is the step-size, $n\beta$ is the effective number of samples that modifies the size of the gradient term in the SGLD step, and $\gamma$ is the localization strength.

| Experiment | § | Dataset | $m$ | $C$ | $T$ | b | $\epsilon$ | $n\beta$ | $\gamma$ |
|---|---|---|---|---|---|---|---|---|---|
| Vision | 4 | ImageNet | 256 | 15 | 1000 | 10 | $1 \times 10^{-4}$ | 10 | 1000 |
| Language | 4 | Pile | 64 | 5 | 1000 | 100 | $8 \times 10^{-7}$ | 2000 | 7000 |
| Scaling | 4 | Pile | 32 | 4 | 500 | 0 | $5 \times 10^{-6}$ | 30 | 300 |
| Retraining ResNet | C | CIFAR10 | 1024 | 4 | 100 | 0 | $1 \times 10^{-5}$ | 200 | 10000 |
| Retraining LLM | C.3 | Wikitext | 128 | 4 | 300 | 150 | $2 \times 10^{-6}$ | 800 | 50000 |
| Language | B | Pile | 32 | 4 | 500 | 0 | $5 \times 10^{-6}$ | 30 | 300 |
| Language | B | Pile | 64 | 5 | 100 | 0 | $5 \times 10^{-5}$ | 30 | 300 |

## B.3 COMPARING THE LOCAL BIF AGAINST EK-FAC

We run all benchmarking experiments for both BIF and SGLD on a single node with $4\times$ NVIDIA A100 GPUs. As given in Table 3, for the BIF estimation, we run SGLD with batch size $m = 32$, number of chains $C = 4$, number of draws per chain $T = 500$, learning rate $\epsilon = 5 \times 10^{-6}$, inverse temperature $n\beta = 30$, and localization strength $\gamma = 300$. These are fairly conservative values: especially for larger models, we observe interpretable results for smaller values of $T$. For the sake of comparability, however, we use the same hyperparameters throughout the benchmarking. Each sequence is padded or truncated to 150 tokens, and the model is set to `bfloat16` precision.

We use the `kronfluence` package for EK-FAC computation (Grosse et al., 2023).[1] This package splits the influence computation into a fit and score step. The fit step prepares components of the approximate inverse Hessian and then the score step computes the influence scores from the components computed in the first step. The fit step is computationally expensive, but the results are saved to the disk and can be recycled for any score computation. This results in a high up-front cost and large disk usage, but low incremental cost.

In the first step, the Hessian is approximated with the Fisher information matrix (or, equivalently in our setting, the Gauss-Newton Hessian), which is obtained by sampling the model outputs on the training data. Since the Pile, which is the dataset used for Pythia training, is too large to iterate over in full, we approximate it by taking a representative subset of $1\,000\,000$ data points, curated using $k$-means clustering (Gao et al., 2021; Kaddour, 2023). Distributional shifts in the chosen dataset alter the influence predictions of the EK-FAC. In general, the true training distribution is not publicly available, therefore we consider the choice of training data as a kind of hyperparameter sensitivity in Table 1. Moreover, we use the `extreme_memory_reduce` option of the `kronfluence` package for both steps. Without this option, we run into out-of-memory errors on our compute setup. Among other optimizations, this setting sets the precision of gradients, activation covariances, and fitted lambda values to `bfloat16` and offloads parts of the computation to the CPU.

The comparison is depicted in Figure 5. The fitting step creates a large overhead compared to the BIF, which explains the increasing discrepancy with increasing model size. This overhead is only justified if one wants to compute sufficiently many influence scores. Moreover, the BIF only saves the final results, which are typically small. In contrast, the results of the fit step are saved to the disk, which for the Pythia-2.8B model occupies 41 GiB.

## B.4 PER-TOKEN INFLUENCE

Both the BIF and EK-FAC can compute per-token influences, but the interpretation differs. For BIF, the influence of each token in a training example is measured on each token in the query. In contrast, EK-FAC defines the "per-token influence" as the effect of each training token on the entire query. We can recover the EK-FAC definition of per-token influence from BIF by summing over

---

[1]The corresponding github repository is available here: `https://github.com/pomonam/kronfluence`

the query tokens. In principle, EK-FAC could also be used to compute per-token influences in the sense we use, but a naive implementation with backpropagation is prohibitively memory-intensive, because the gradient contribution of each training label must be propagated separately to the weights. Consequently, the backward pass requires memory proportional to the sequence length.

## C RETRAINING EXPERIMENTS

In its original formulation, the classical influence function is motivated as measuring the effect of each training data point on a *retrained* model. That is, for each $\mathbf{z}_i \in \mathcal{D}_{\text{train}}$, if the model is retrained from initialization on the leave-one-out dataset $\mathcal{D}_{\text{train}} \setminus \{\mathbf{z}_i\}$, what is the effect on the observable $\phi$?

### C.1 LINEAR DATAMODELLING SCORE

Both classical and Bayesian influence functions approximate the effect of $\mathbf{z}_i$'s exclusion from $\mathcal{D}_{\text{train}}$ as *linear*. That is, given a subset $\mathcal{D} \subseteq \mathcal{D}_{\text{train}}$, write $\phi(\mathcal{D})$ as the value of the observable $\phi$ corresponding to a model trained on $\mathcal{D}$:

$$\phi_{\text{C}}(\mathcal{D}) := \phi(\boldsymbol{w}^*(\mathcal{D})), \qquad \boldsymbol{w}^*(\mathcal{D}) \in \arg\min_{\boldsymbol{w} \in \mathcal{W}} \sum_{\mathbf{z}_i \in \mathcal{D}} \ell_i(\boldsymbol{w}).$$

in the classical perspective and

$$\phi_{\text{B}}(\mathcal{D}) := \mathbb{E}_{\boldsymbol{w} \sim p(\boldsymbol{w}|\mathcal{D})}[\phi(\boldsymbol{w})]$$

in the Bayesian perspective. In either case, we approximate $\phi(\mathcal{D})$ as linear in the set $\mathcal{D}$:

$$\phi(\mathcal{D}) \approx \sum_{i=1}^{n} \tau_i [\mathbf{z}_i \in \mathcal{D}],$$

where each $\tau_i \in \mathbb{R}$ is a training data attribution measure associated to $\mathbf{z}_i$ and $\phi$, e.g. $\text{IF}(\mathbf{z}_i, \phi)$ or $\text{BIF}(\mathbf{z}_i, \phi)$.

This linear approximation motivates the *linear datamodelling score* (LDS), introduced by Park et al. (2023b). Given the training dataset $\mathcal{D}_{\text{train}}$ of cardinality $n$ and a query set $\mathcal{D}_{\text{query}}$, we let the query losses $(\phi_{\mathbf{z}_j} = \ell(\mathbf{z}_j; -))_{\mathbf{z}_j \in \mathcal{D}_{\text{query}}}$ be our observables and suppose we are given TDA measures $(\boldsymbol{\tau}_{\mathbf{z}_j})_{\mathbf{z}_j \in \mathcal{D}_{\text{query}}}$, with each $\boldsymbol{\tau}_{\mathbf{z}_j} \in \mathbb{R}^n$. To measure the LDS of $(\boldsymbol{\tau}_{\mathbf{z}_j})_{\mathbf{z}_j}$, we subsample datasets $\{\mathcal{D}_k\}_{k=1}^{K}$ with each $\mathbf{z}_i \in \mathcal{D}_k$ with probability $\alpha_{\text{retrain}} \in \{0.1, 0.3, 0.5, 0.7\}$ iid. (For our experiments, we set $K = 100$). The LDS of $(\boldsymbol{\tau}_{\mathbf{z}_j})_{\mathbf{z}_j}$ is then the average over $1 \le k \le K$ of the correlation between the true retrained observable and the linear approximation from $(\boldsymbol{\tau}_{\mathbf{z}_j})_{\mathbf{z}_j}$:

$$\text{LDS}((\boldsymbol{\tau}_{\mathbf{z}_j})_{\mathbf{z}_j \in \mathcal{D}_{\text{query}}}; (\phi_{\mathbf{z}_j})_{\mathbf{z}_j \in \mathcal{D}_{\text{query}}}, \{\mathcal{D}_k\}_{k=1}^{K})$$

$$= \frac{1}{K} \sum_{k=1}^{K} \rho_{\text{s}} \left( (\phi_{\text{C},\mathbf{z}_j}(\mathcal{D}_k))_{\mathbf{z}_j \in \mathcal{D}_{\text{query}}}, \left( \sum_{i=1}^{n} \tau_{\mathbf{z}_j, i}[\mathbf{z}_i \in \mathcal{D}_k] \right)_{\mathbf{z}_j \in \mathcal{D}_{\text{query}}} \right), \quad (19)$$

where $\rho_{\text{s}}$ is Spearman's rank correlation coefficient. Each $\phi_{\text{C},\mathbf{z}_j}(\mathcal{D}_k)$ is computed by retraining the model on $\mathcal{D}_k$ and evaluating the loss on $\mathbf{z}_j$. Note that, regardless of whether we evaluate the LDS of an approximate classical IF or the BIF, we use the classical version of the retrained observable $\phi_{\text{C}}$. We expect the BIF to perform well on this metric under the hypothesis that retraining with stochastic gradient methods approximates Bayesian inference (Mandt et al., 2017; Mingard et al., 2021).

### C.2 LDS EXPERIMENT DETAILS AND RESULTS

We evaluate the LDS of the EK-FAC, BIF, GradSim, TRAK on a ResNet-9 model with $1\,972\,792$ parameters (He et al., 2015) trained on the CIFAR-10 (Krizhevsky, 2009) image classification dataset. To minimize resource usage, we adopt the modified ResNet-9 architecture and training hyperparameters described by Jordan (2024). In addition, we set aside a warmup set $\mathcal{D}_{\text{warmup}}$ of 2500 images. Before the actual training runs, we perform a short warmup phase on $\mathcal{D}_{\text{warmup}}$ to prime the optimizer state. The training hyperparameters are summarized in Table 4.

| Hyperparameter | Image Classification | Word Prediction (NLP) |
|---|---|---|
| Training algorithm | SGD | AdamW |
| Epochs | 1 (8) | 1 |
| Batch size | 1024 | 256 |
| Momentum | 0.85 | $\beta_1 = 0.9$ |
| Weight decay | 0.0153 | 0.01 |
| Learning rate | 10.0 | $3 \times 10^{-5}$ |
| Warmup steps | 100 | — |
| Label smoothing | 0.2 | 0.0 |
| Bias scaler | 64.0 | — |
| Whiten bias epochs | False | False |
| Gradient accumulation steps | 1 | 1 |

Table 4: Training hyperparameters for retraining experiments. The foundational ResNet-9 model used to compute TDA scores was trained for 8 epochs. The retrained image-classification models were trained for a single epoch. For the next-token-prediction task, we used the pretrained Pythia-14m model at checkpoint 45,000.

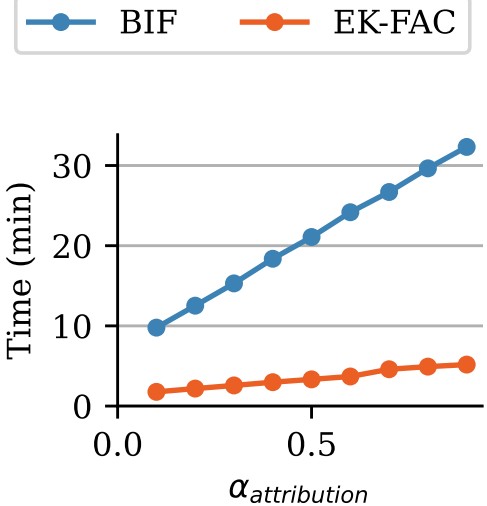

Figure 6: **Wall-clock time** as a function of $\alpha_{\text{attribution}}$ for BIF and EK-FAC in the retraining experiments. Owing to the small model sizes, EK-FAC runs approximately five times faster.

As described in Appendix C.1, we evaluate LDS by re-training the ResNet-9 100 times from initialization on random subsamples of the full CIFAR-10 training set, excluding the warmup set ($n = 47\,500$ images). Each subsample contains $n_{\text{retrain}} = \alpha_{retrain}\alpha_{attribution}n$ images. We then use the full test set ($q = 10\,000$ images) as the query set, i.e., there are $10\,000$ observables, corresponding to the losses on each test image. Thus, both EK-FAC and BIF TDA scores comprise a $n_{\text{attribution}} \times 10\,000$ matrix. The hyperparameters for the SGLD estimation of the BIF are given in Table 3. For EK-FAC, we set the dampening factor to $10^{-8}$. Both TDA techniques are computed on a single model checkpoint trained with the hyperparameters listed in Table 4. Figure 6 displays the wall-clock times of the BIF and EK-FAC computation. In these experiments, EK-FAC is around five times faster than the BIF. This advantage is largely due to the small model sizes ($\sim 2 \times 10^6$ parameters), which results in a short fitting stage.

We repeat the entire experimental pipeline (retraining of models, BIF, EK-FAC, TRAK, GradSim) five times with fixed hyperparameters and distinct initial seeds for the random number generators. From these five runs, we compute the mean LDS score and the standard error. The LDS scores of each individual run are displayed in Figure 7. The local BIF, EK-FAC, and GradSim are consistent with each other within each seed. However, the LDS score varies substantially across seeds. This

suggests either that the LDS score is not a reliable quantitative measure for evaluating TDA methods, or that influence functions in general do not capture the true counterfactual impact of individual training examples.

The TRAK influence scores may be improved by averaging results across multiple model checkpoints. Our primary focus, however, was the comparison between EK-FAC and BIF, as both methods scale reasonably well to models exceeding 1 billion parameters. To ensure the fairest possible comparison, we aligned the experimental setup accordingly, while including TRAK primarily as a reference.

Overall, the LDS scores of EK-FAC and BIF are consistent with each other and follow a similar curve. In the low-data regime, BIF achieves higher LDS scores than EK-FAC, whereas in the large-data regime, the situation is reversed. As we show in Appendix A.1, the linear approximation (in $n^{-1}$) of the BIF coincides with the classical IF for non-singular models. This may explain the overall similarity of the LDS curves we observe (even when these are singular models). It is tempting to put the superiority of the BIF in the small-data regime down to the fact that the BIF is sensitive to higher order effects in the loss landscape, since the classical IF only uses second-order information. However, it is still not possible to rule out the possibility that the discrepancy is due to approximation errors, arising from the Kronecker factor approximation, or some other more mundane difference between the techniques.

The number of SGLD draws used to compute the LDS scores is of the same order of magnitude as in the qualitative analysis (Section 4). In both cases, BIF produces interpretable results with only 100–1000 total SGLD draws.

### C.3 LINEAR DATAMODELING SCORES ON FINETUNING EXPERIMENTS

Table 5: Linear datamodeling Score for Pythia-14M model on the Wikitext finetuning task. We used $\alpha_{\text{retrain}} = 0.5$ and 100 subsamples.

| Method | LDS | Std. Dev. |
|---|---|---|
| GradSim | 0.198 | 0.109 |
| EK-FAC | 0.221 | 0.125 |
| Local BIF | 0.110 | 0.112 |

The LDS score also serves as a benchmark for the local Bayesian influence function (BIF) for language models in next-token prediction tasks. To this end, we finetune a pretrained Pythia-14M model at checkpoint 45,000 on 5,680 `Wikipedia (en)` text samples filtered from the Pile (Gao et al., 2021). The TDA methods predict the per-sample loss on an evaluation set, consisting of 385 English Wikipedia text samples. Each text example is tokenized and wrapped into contexts of fixed length of 1024 tokens. After this preprocessing step, the final dataset contains 4,096 attribution sequences and 256 evaluation sequences.

We perform full-parameter finetuning of the pretrained Pythia-14M model for one epoch using AdamW with a linear learning rate schedule. The training hyperparameters are listed in Table 4. In total, we train 100 models using random subsamples of the full attribution set with $\alpha_{retrain} = 0.5$. Afterwards, we compute per-sequence evaluation losses by averaging token-level negative log-likelihoods across each sequence.

The row "Retraining LLM" in Table 3 lists the SGLD hyperparameters. The per-sequence losses on the evaluation set serve as the measurement function for both the BIF and EK-FAC. We compute EK-FAC influence scores by fitting Kronecker-factored curvature components on the same finetuning corpus. All experiments are performed on a node with 4 NVIDIA A100 GPUs (80 GB VRAM), and both methods use identical finetuned checkpoints and evaluation losses.

We compute the EK-FAC and the GradSim TDA scores with the `kronfluence` package by setting the strategy option to `ek-fac` for the former and to `identity` for the latter. The resulting LDS scores are summarized in Table 5.

We do not report LDS scores for TRAK because, to the best of our knowledge, there is currently no publicly available implementation of TRAK for large language models.

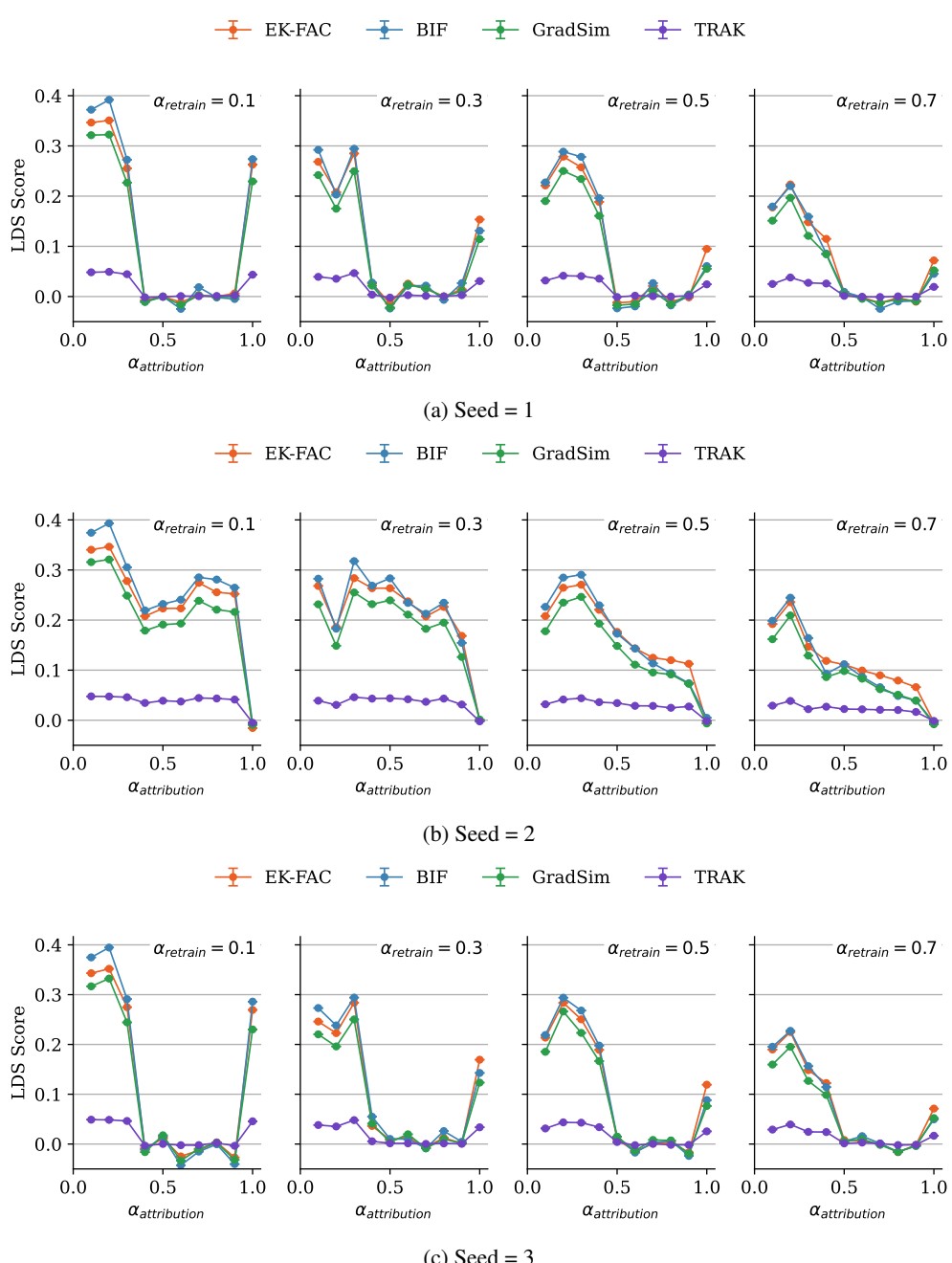

Figure 7: **Individual LDS values across different seeds.** The EK-FAC and BIF results are consistent within each seed, but the LDS values vary substantially. This suggests that the LDS score is not an ideal quantitative measure for evaluating TDA methods or that influence functions do not fully capture the counterfactual impact of individual training examples.

We consider two approaches for deriving per-sequence TDA scores from the per-token losses. The measurement dataset comprises the attribution and evaluation sets, yielding $n_{\text{measurement}} = n_{\text{attribution}} + n_{\text{query}}$ text samples. Consequently, the SGLD loss trace tensor has the shape $(C, T, n_{\text{measurement}}, S)$.

First, we stack the tensor along the chain axis, resulting in a tensor of shape $(C \times T, n_{\text{measurement}}, S)$. This is equivalent to treating all draws as belonging to a single chain. This approximation is justified

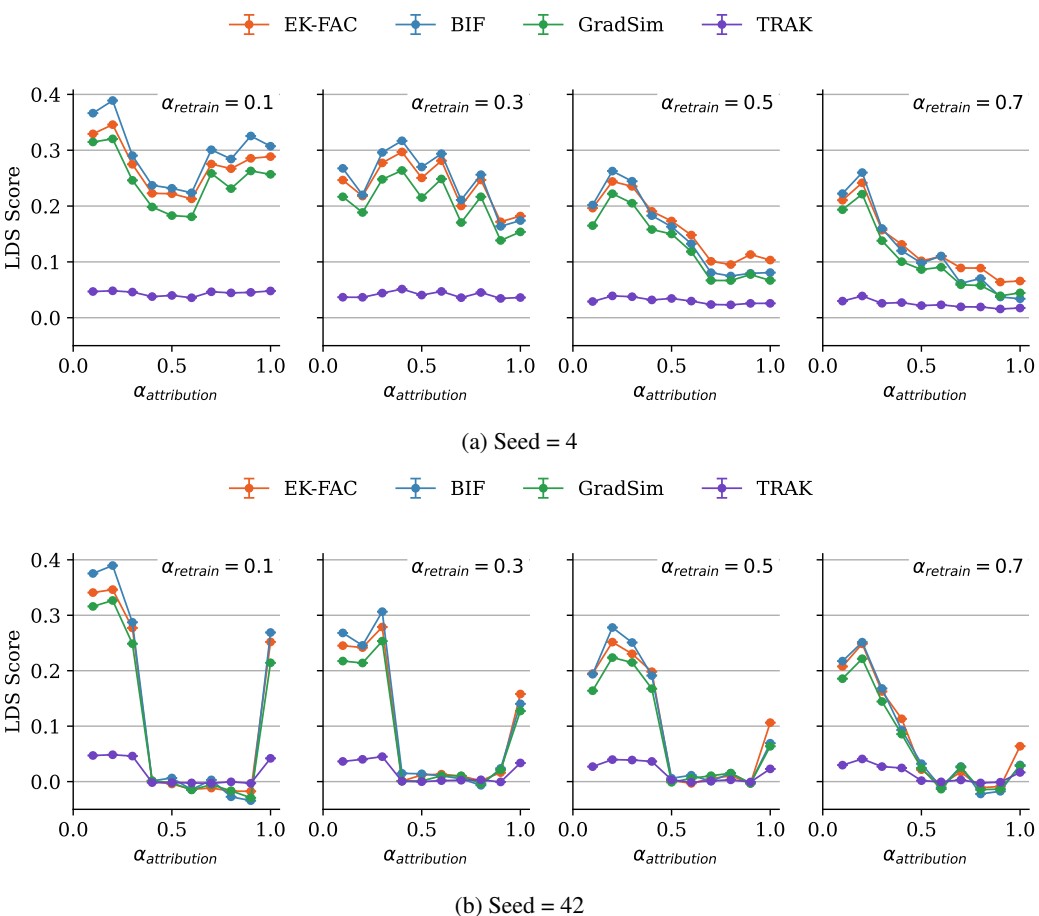

(a) Seed = 4

(b) Seed = 42

Figure 8: **Individual LDS scores across different seeds.** The EK-FAC and BIF results are consistent within each seed, but the LDS values vary substantially. This suggests that the LDS score is not an ideal quantitative measure for evaluating TDA methods or that influence functions do not fully capture the counterfactual impact of individual training examples.

when the SGLD sample set contains substantially more samples than $C \times T$, since the probability of duplicate samples is then negligible.

**Implementation A (sequence-level).** We average the per-token loss along the token dimension, which reduces the tensor from shape $(C \times T, n_{\text{measurement}}, S)$ to $(C \times T, n_{\text{measurement}})$. Next, we evaluate the BIF correlation matrix, obtaining an array of dimension $(n_{\text{measurement}}, n_{\text{measurement}})$. Finally, we extract the top-right block of this BIF correlation matrix, with shape $(n_{\text{attribution}}, n_{\text{query}})$, for subsequent analysis.

**Implementation B (token-level).** In this variant, we first compute $n_{\text{attribution}} \times n_{\text{query}}$ BIF covariance matrices, each of dimension $(S, S)$. We then summarize each covariance matrix by averaging its entries, reducing it from an $(S, S)$ matrix to a scalar. We subsequently assemble these scalars into a final matrix of shape $(n_{\text{attribution}}, n_{\text{query}})$.

Both ways result in the TDA prediction $\boldsymbol{\tau}_{\mathbf{z}_j}$ from which we compute the LDS score in Equation (19). In practice, both implementations yield the same LDS scores. Therefore, we default to the sequence method because it is orders of magnitude faster.

The BIF LDS score is hyperparameter-sensitive in the language model setting. See Figure 14 for a hyperparameter sweep. The LDS score of 0.11 reported in Table 5 corresponds to the best LDS score after a wide hyperparameter search. The hyperparameter dependence of the BIF poses a bottleneck

that will be addressed in upcoming work. We expect that a systematic understanding of SGLD hyperparameters is essential to close the current gap between the BIF LDS score and the ones from EK-FAC and GradSim.

The BIF LDS score is hyperparameter-sensitive in the language model setting. See Figure 14 for a hyperparameter sweep. The LDS score of 0.11 reported in Table 5 corresponds to the best LDS score after a wide hyperparameter search. We believe this gap is not fundamental to the BIF framework but reflects the current immaturity of SGLD sampling practice: the interplay between the BIF's inverse temperature ($\beta$), localization strength ($\gamma$), and step size ($\epsilon$) in the language model regime is not yet well characterized. Improving samplers and developing a principled understanding of hyperparameter selection is a central focus of ongoing work, and we expect that this work will close the current gap.

### C.4 SGLD HYPERPARAMETERS

We analyzed the dependence on the SGLD hyperparameters by sweeping over $(b, n\beta, \gamma) \in [0, 100] \times [100, 300, 1, 000, 3, 000] \times [1, 000, 3, 000, 10, 000, 30, 000, 100, 000]$, using $\alpha_{\text{attribution}} = 0.1$ and computing the corresponding LDS scores. The grid plots Figure 9–Figure 12 show the resulting loss traces and LDS scores for $\alpha_{\text{retrain}} = 0.1$ and $\alpha_{\text{retrain}} = 0.3$. These comparisons indicate that for $b = 100$, the LDS scores remain stable across hyperparameter choices as long as the loss trace converges. Furthermore, Figure 13 demonstrates that this stability holds independently of the choice of $\alpha_{\text{retrain}}$.

## D ADDITIONAL QUALITATIVE RESULTS

### D.1 BIF AND EK-FAC ON VISION

See Figure 15 for additional qualitative comparisons between BIF and EK-FAC for the Inception-v1 image classification model (Szegedy et al., 2015) on ImageNet data (Deng et al., 2009). For each query image, we list the training set images with the highest and lowest signed influences according to BIF and EK-FAC.

**Interpreting high-influence samples.** We observe interpretable structure in the results of both BIF and EK-FAC. The highest-influence training images for each query image are often visually similar images with the same label—intuitively, correctly-labeled training examples of, for instance, a fox terrier (Figure 15, row 3), should help the model better identify fox terriers in the query set. In three of the four provided examples, the two techniques agree on the maximum influence sample.

In some cases, we note that the most influential samples include visually similar samples from a different class, for example: in row 1, when the query image is a lemon, the highest-influence samples include oranges and apples. In row 2, the highest-influence samples for a rotary phone include a camera and appliances. Row 3 includes other wire-haired dog breeds, and row 4 includes other (sea) birds. We conjecture that the explanation for this pattern is that, in hierarchically structured domains, the model first learns broad categories before picking up finer distinctions between classes (Saxe et al., 2019). Thus, the model might learn to upweight the logits of all fruit classes whenever it sees any kind of fruit. Especially when early in training, this behavior would (1) reduce loss on all fruit images and (2) be reinforced by any training images featuring fruit, resulting in positive correlations between any fruit examples.

**Interpreting low-influence samples.** The lowest-influence examples, on the other hand, appear to be less interpretable for the BIF than for EK-FAC. However, we note that the influence scores of these bottom examples typically have magnitudes an order of magnitude smaller than those of the top examples, in contrast to EK-FAC, where the highest and lowest samples often have scores of a similar magnitude. Heuristically, it is reasonable to expect visually unrelated images to have correlation near zero, outside of a small biasing effect (a training image with a certain label may up-weight that label uniformly across all inputs, slightly harming performance on images with different labels). Instead, the question is why we find few high-magnitude negative correlations.

**Disagreement between highest- and lowest-influence samples.** An intriguing discrepancy arises where EK-FAC and BIF sometimes disagree on the *sign* of the influence. For instance, in row 1 of Figure 15, images of oranges have negative influence (positive correlation) according to BIF, yet

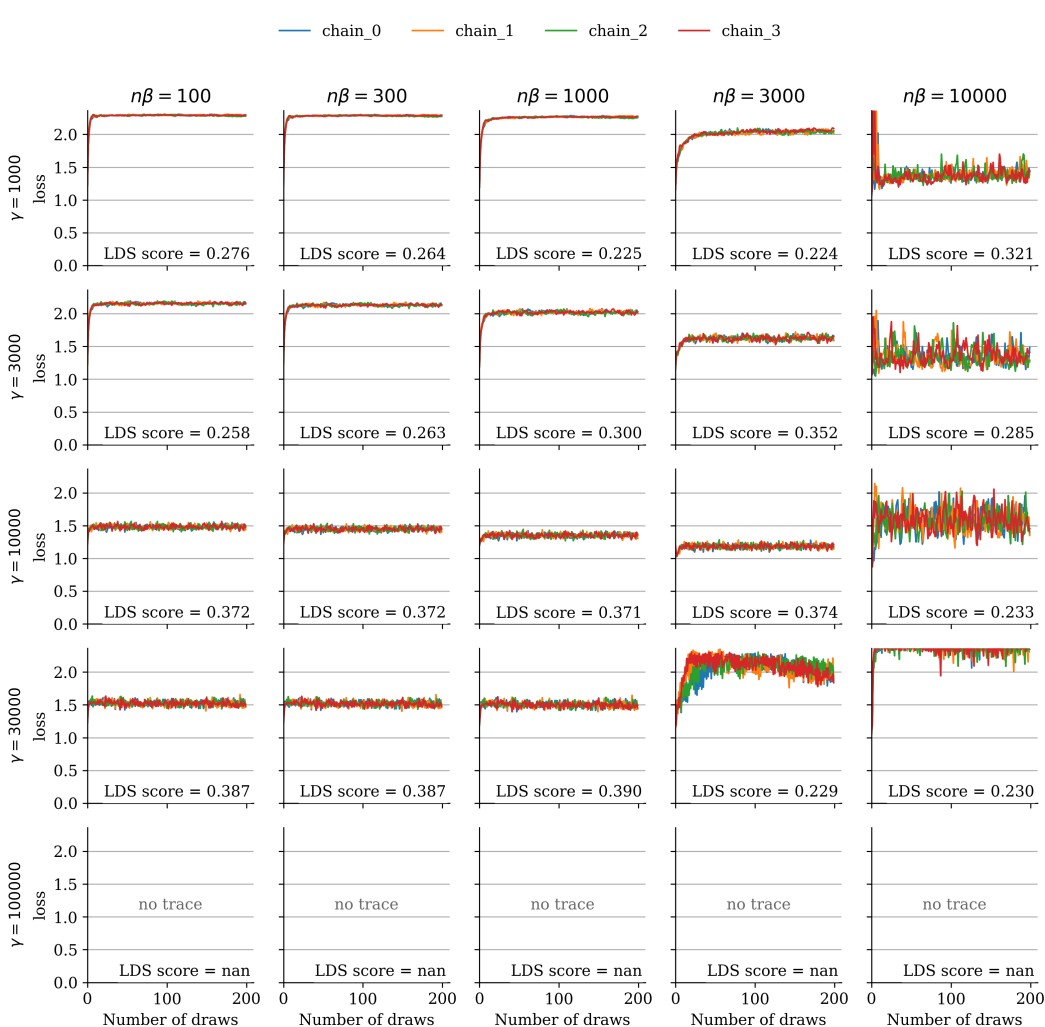

Figure 9: **Loss traces and LDS scores for** $b = 0$ **and** $\alpha_{\mathbf{retrain}} = 0.1$. NaNs mark divergent SGLD estimates that failed to converge.

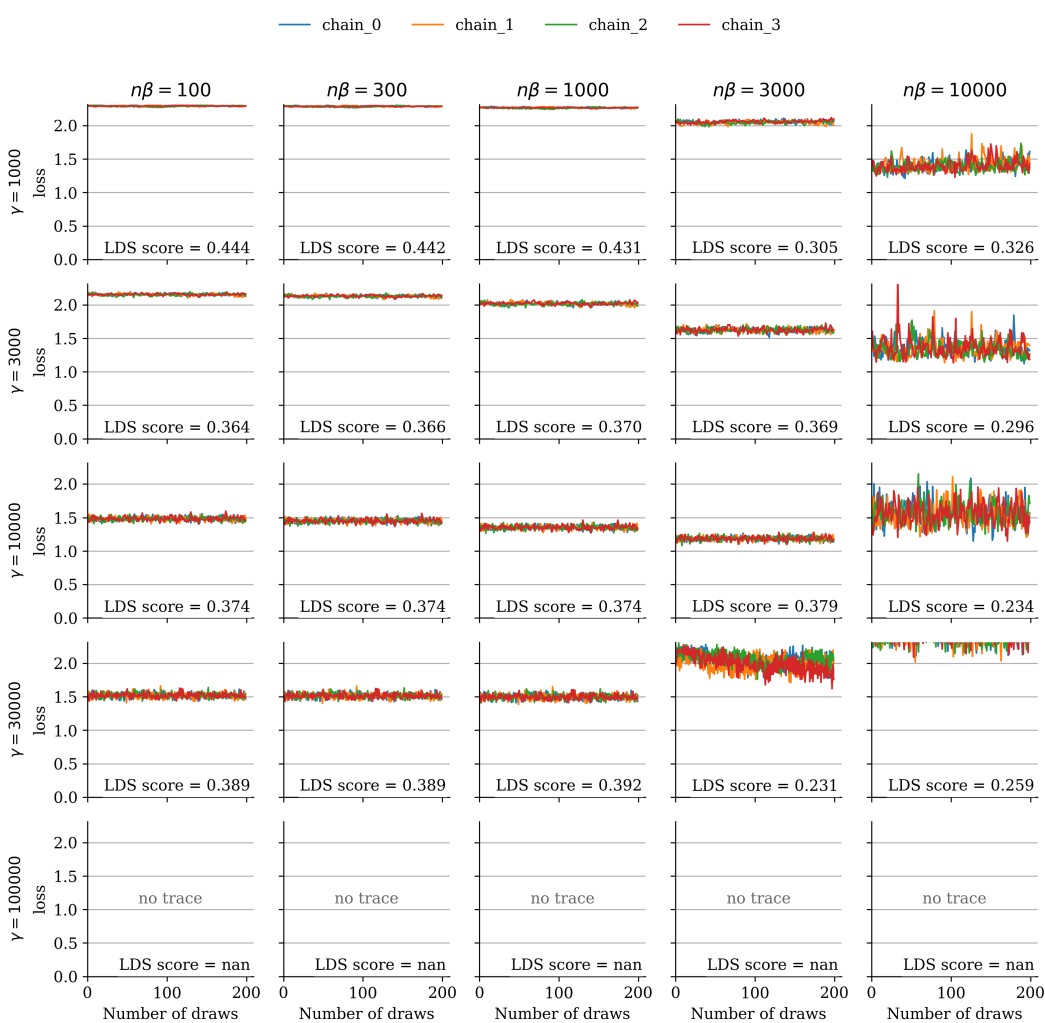

Figure 10: **Loss traces and LDS scores for** $b = 100$ **and** $\alpha_{\text{retrain}} = 0.1$. NaNs mark divergent SGLD estimates that failed to converge.

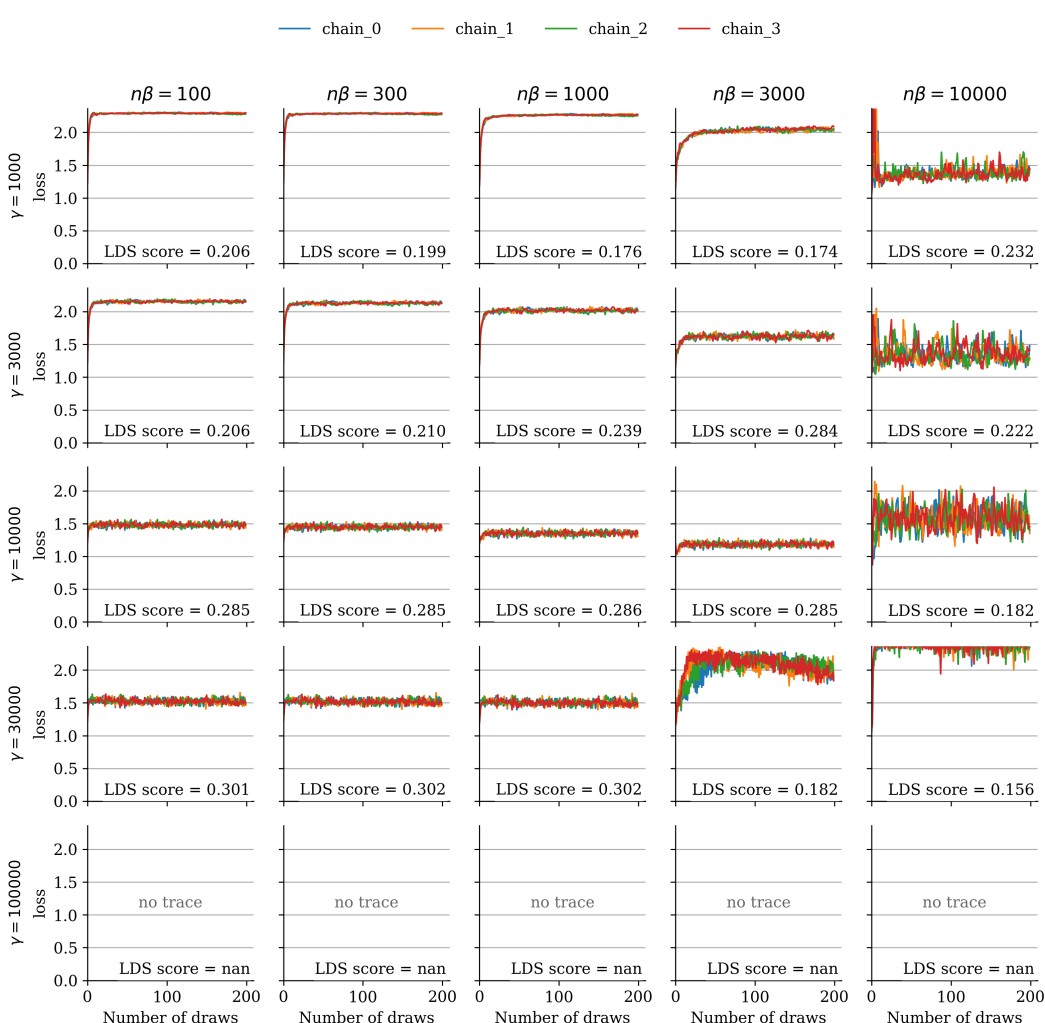

Figure 11: **Loss traces and LDS scores for** $b = 0$ **and** $\alpha_{\text{retrain}} = 0.3$. NaNs mark divergent SGLD estimates that failed to converge.

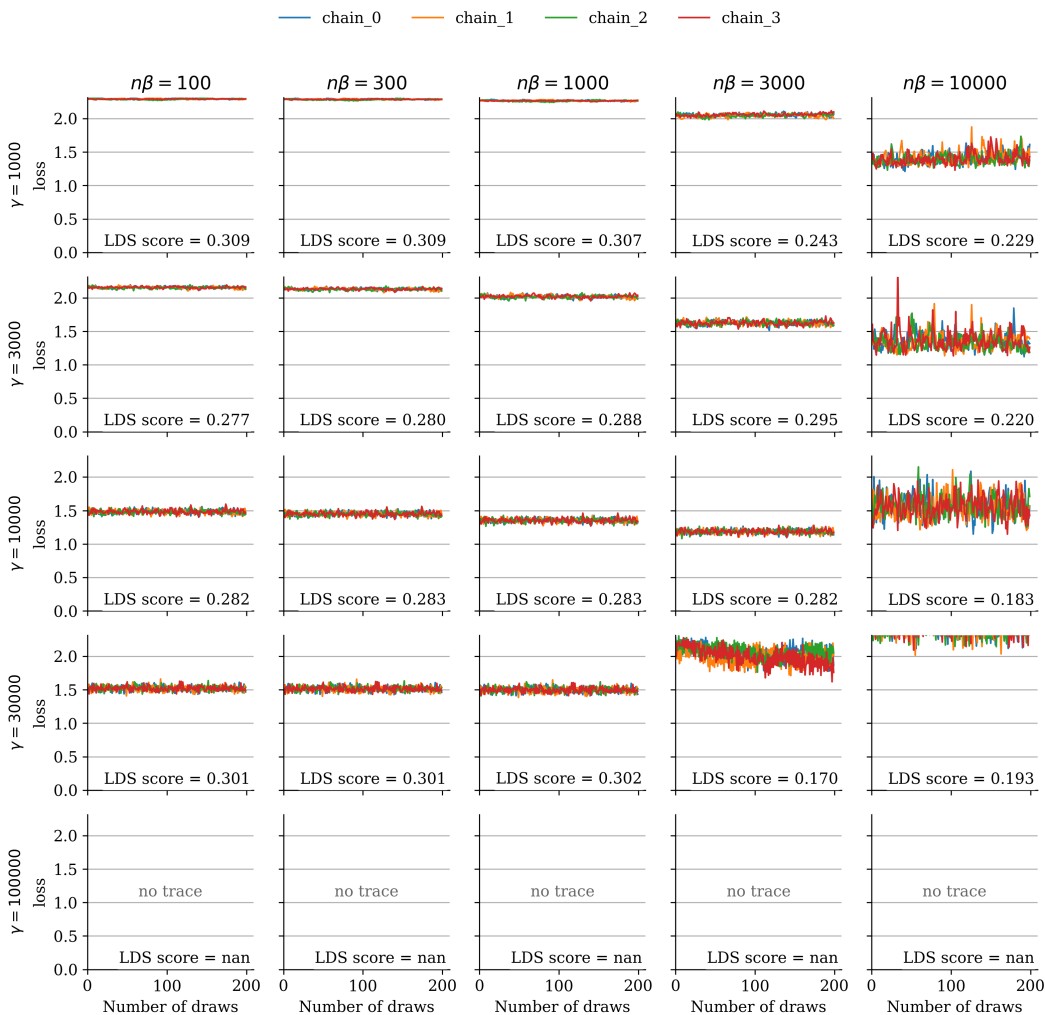

Figure 12: **Loss traces and LDS scores for** $b = 100$ **and** $\alpha_{\mathbf{retrain}} = 0.3$. NaNs mark divergent SGLD estimates that failed to converge.

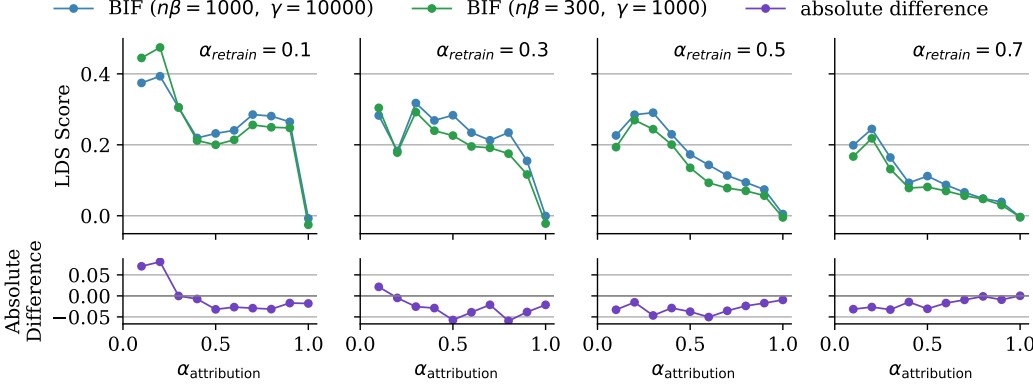

Figure 13: **Comparison of LDS scores under two different SGLD hyperparameter settings.** The lower panel shows the absolute difference between LDS scores. Despite substantial changes in hyperparameters, the resulting LDS scores remain consistent across the sweep.

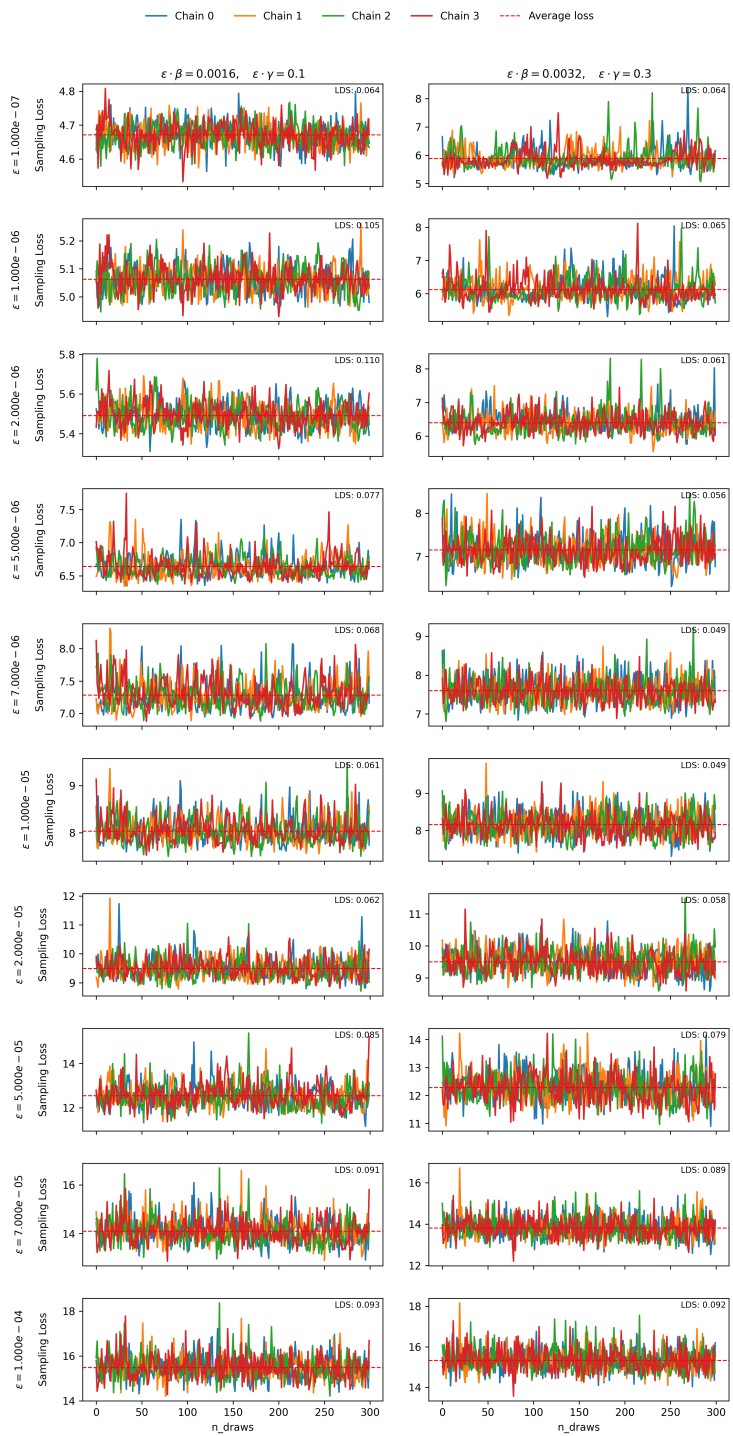

Figure 14: Loss traces and LDS scores in the language-model setup. The remaining SGLD hyperparameters are: $C = 4$, $m = 128$, $T = 300$, $b = 150$, and $\alpha_{\text{retrain}} = 0.5$.

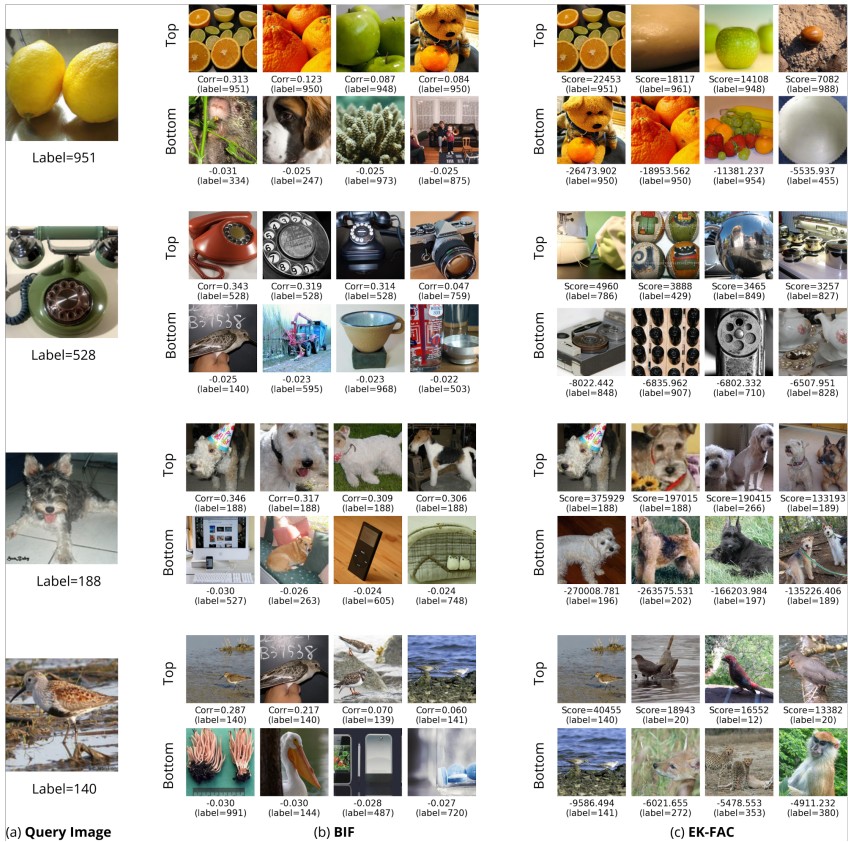

Figure 15: **BIF vs. EK-FAC for Inception-v1 on ImageNet.** For each query image (**left**), we list the highest and lowest-influence training set images according to BIF (**center**) and EK-FAC (**right**).

positive according to EK-FAC; a similar reversal is observed in the bottom row. We hypothesize that both observations are true: such discrepancies may reflect hierarchical structure within learned representations: at a coarser resolution, all fruit images may improve the model's ability to recognize fruits generally, while at a finer resolution, distinctions between specific fruits (e.g., lemons vs. oranges) introduce negative correlations. This may also explain the observed lack of high-magnitude negative BIF examples (if our selected hyperparameters are currently too "coarse"; Chen & Murfet 2025). Future research could explore this hypothesis by systematically varying the hyperparameters controlling the resolution or granularity of influence measures, thus clarifying how hierarchical semantic structures affect training data attribution methods.

## D.2   BIF AND EK-FAC ON LANGUAGE

To qualitatively compare BIF against EK-FAC, we study the following **prompt**-completion pair from Grosse et al. (2023):

**Human:  What is your objective function?  Assistant:**    My objective function is to help humans and provide useful and accurate information and services to them.  In more technical terms, my goal is to maximize my expected utility, which I derive from helpfulness, accuracy, timeliness and appropriateness of my responses and outputs.  Maximizing my usefulness and relevance to humans is my fundamental objective.  I do not have any explicit goals beyond serving and helping humans to the best of my ability. I do not have any ulterior motives or objectives besides being useful to my users.

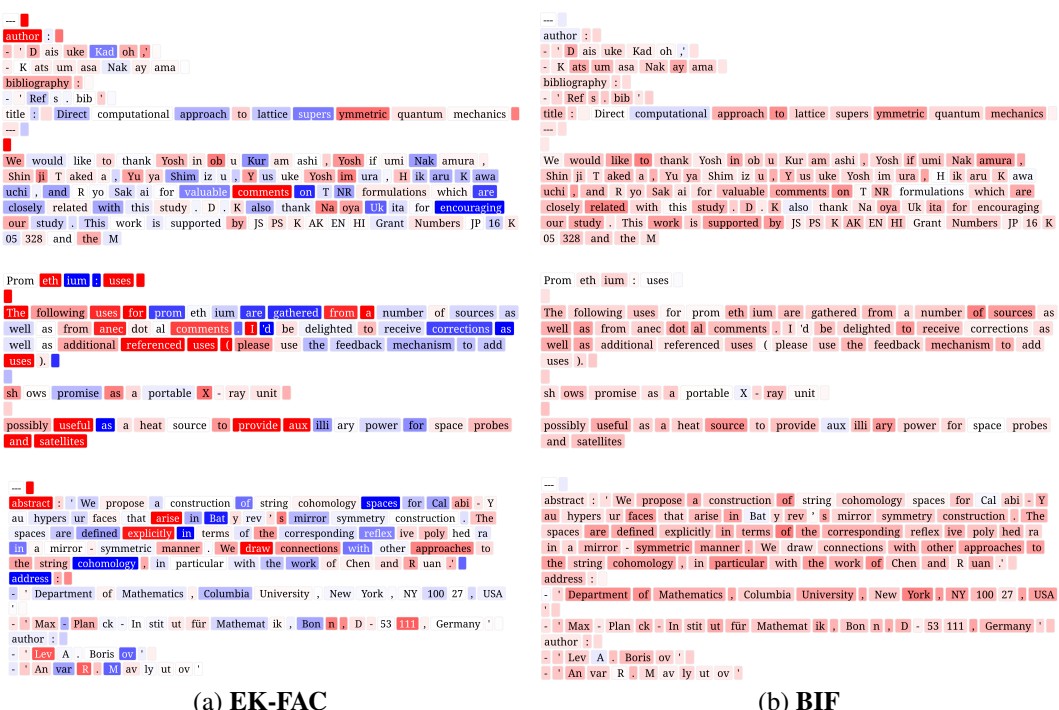

(a) **EK-FAC**                    (b) **BIF**

Figure 16: **EK-FAC vs. BIF on Pythia 2.8B.** The query is the completion "My objective function is..." in the prompt-completion pair in Appendix D.2. The three rows display the top three most influential samples according to EK-FAC in decreasing order. Tokens are colored by their EK-FAC score (left) or BIF (right).

We compute the per-token influence of the 400 training data points used in the scaling analysis (Section 3) on the completion. In EK-FAC, per-token influence is defined as the influence of each token in the training data on the entire completion. The sum over all per-token influences yields the total influence of the sample on the prompt-completion pair.

**Both EK-FAC and BIF perform poorly on Pythia-2.8B.**   For Pythia 2.8B, we show the three most influential samples according to EK-FAC in Figure 16 and the three most influential samples according to the BIF in Figure 17. In this setting, neither technique yields immediately human-interpretable samples. Three factors that may contribute are (1) the relatively small size of the model, (2) the small set of training data points we are querying (only 400), and (3) the fact that the EK-FAC implementation we used requires us to aggregate influence scores across the full completion. As we show in Appendix D.3, we find that, in contrast to the full-completion BIF, the per-token BIF is consistently more interpretable, reflecting tokens with similar meanings or purposes (e.g., countries, years, numbers, jargon, same part of speech).

**Token overlap accounts for much of the influence in small models.**   Grosse et al. (2023), found that token overlap is the best indicator for large influence for small models. For larger models, this changes to more abstract similarities. With the BIF, Figure 17 suggests the same result: the most influential samples are those that have a large token overlap between the sample and the completion. For example, the `.` tokens correlate strongly and appear often on both sides. Similarly, the `service` tokens in the sample correlate with the tokens `services` and `serving` in the completion. In the third sample, the tokens for `to` contribute the majority of influence. Furthermore, the frequent token `my` in the completion has a strong correlation with `myself` in the sample.

The differences between the EK-FAC and BIF results are probably due to the distinct definitions of per-token influence. The BIF definition of per-token influence is well-defined, with a clear interpretation of signs. Furthermore, repeating the EK-FAC computation with the same settings

sometimes leads to different results. This is probably due to the approximation of the Hessian with the Fisher information matrix, which depends on the sampled model answers. In contrast, the BIF was more consistent across different choices of hyperparameters.

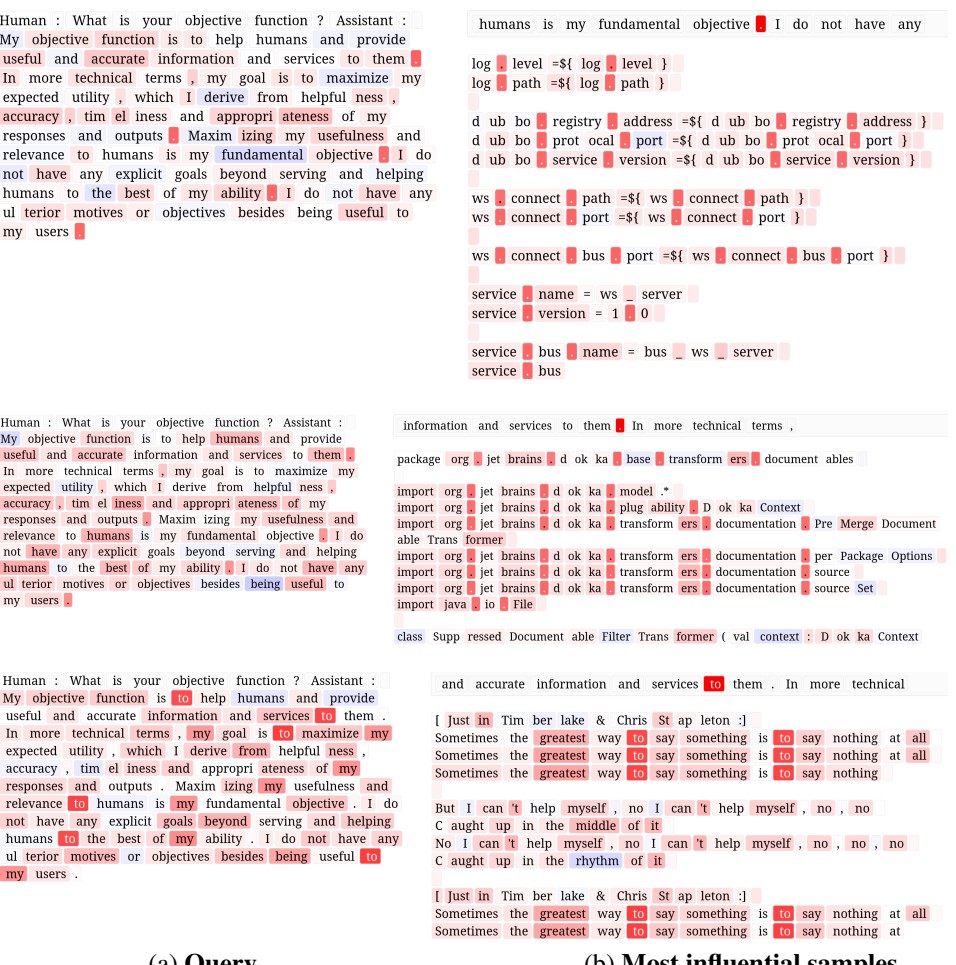

(a) **Query**                    (b) **Most influential samples**

Figure 17: **Most influential samples according to BIF.** The query is the completion "My objective function is..." in the prompt-completion pair in Appendix D.2. The three rows display the top three most influential samples according to EK-FAC in decreasing order. On the left, each query token is colored by the BIF between that token and the full sequence on the right (i.e., summed over all tokens). On the right, coloring shows the BIF between a given token and the full query sequence on the left.

## D.3    Per-token BIF for Pythia 2.8B and 14M

Here we show additional examples for the per-token BIF on Pythia 2.8B (Figure 18) and Pythia 14M (Figures 19 and 20).

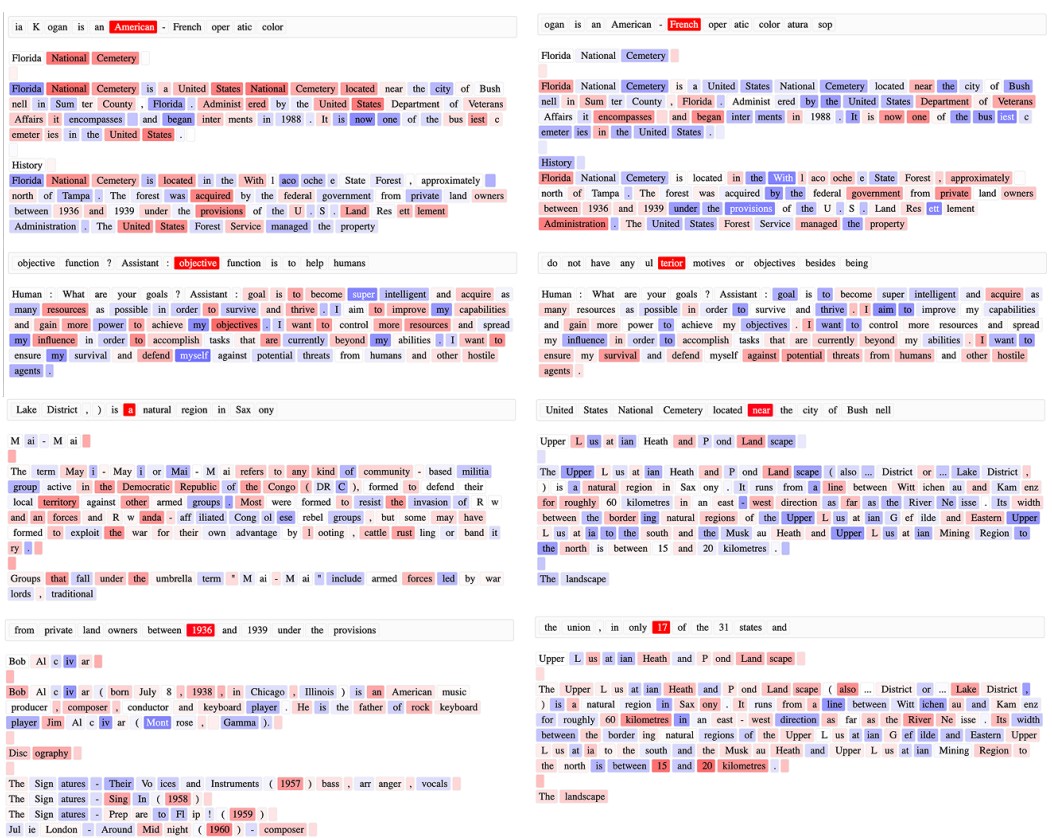

Figure 18: **Additional results for per-token BIF on Pythia-2.8B.**

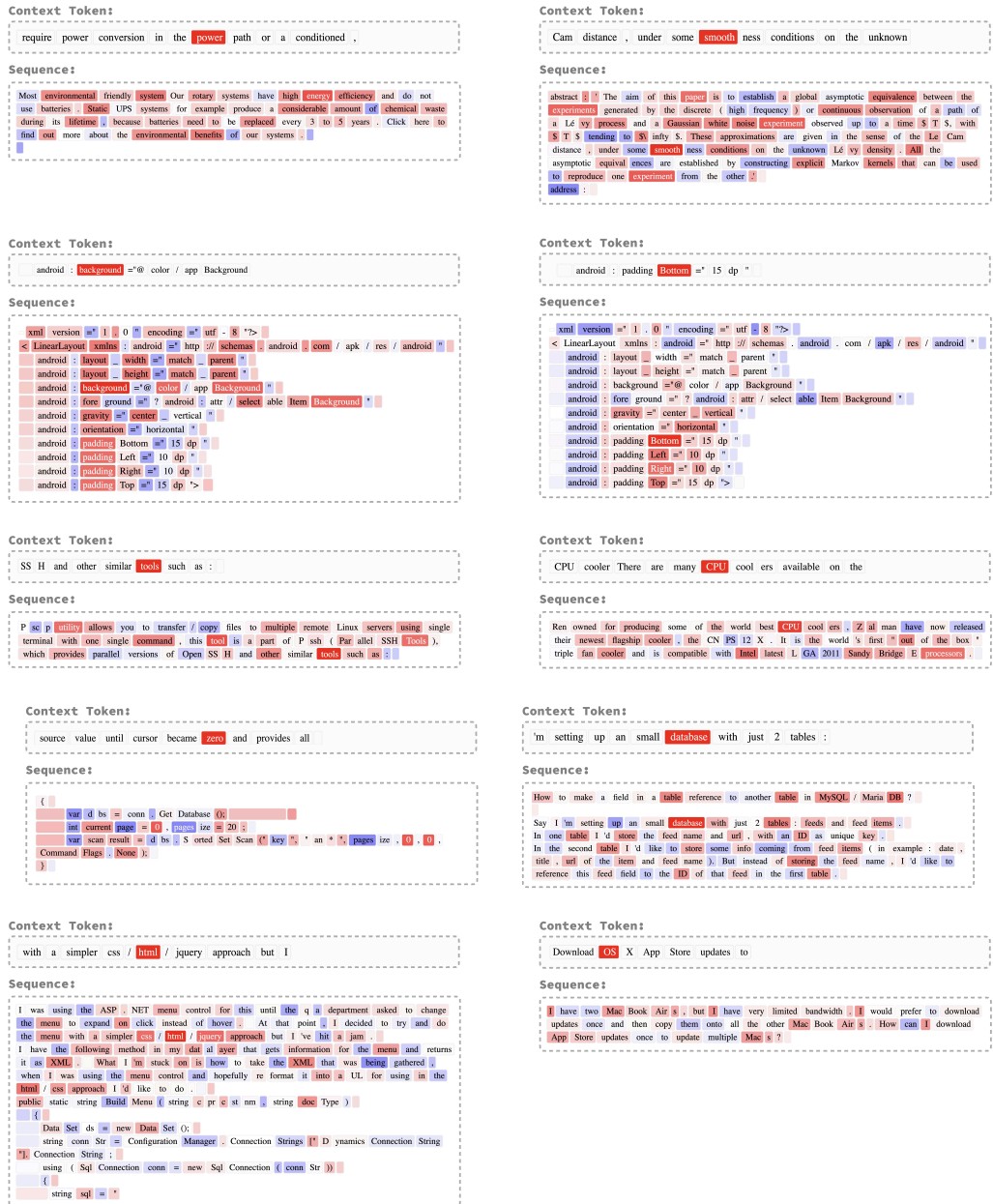

Figure 19: **Additional results for per-token BIF on Pythia 14M.**

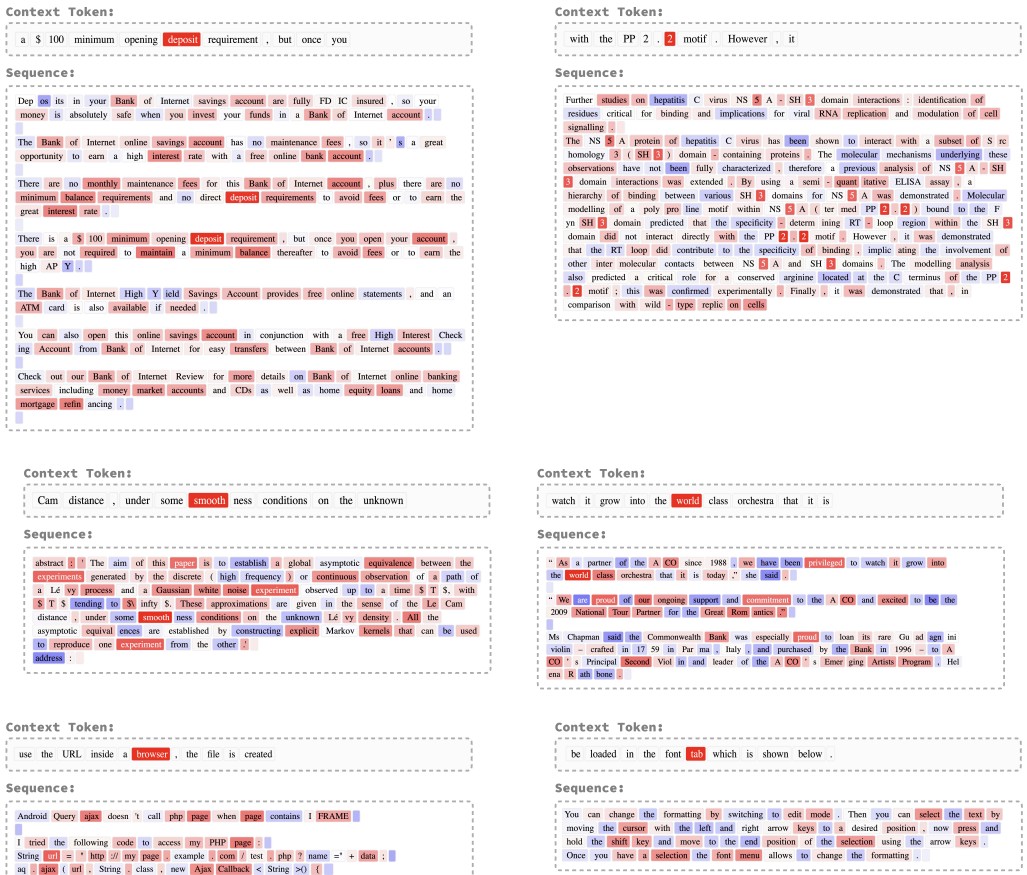

Figure 20: **Additional results for per-token BIF on Pythia 14M.**

