# OpenReview forum: "Bayesian Influence Functions for Hessian-Free Data Attribution"
_ICLR.cc/2026/Conference — ICLR 2026 Poster_

### Official Review · Reviewer_UhDc · 2025-10-26

**Soundness:** 3
**Presentation:** 3
**Contribution:** 3
**Rating:** 4
**Confidence:** 3

**Summary:**

This paper proposes a new method BIF to analyze how individual training samples affect model behavior without requiring Hessian inversion. Classical influence functions break down for deep neural networks because their loss landscapes are highly singular and BIF addresses this by replacing Hessian based inverses with covariance estimates computed over a localized Bayesian posterior using stochastic gradient MCMC sampling. This approach defines influence as the negative covariance between a sample’s loss and an observable, capturing higher-order geometric effects in the loss landscape that classical IF miss. The authors provide a scalable implementation that can handle billion-parameter models, demonstrating strong agreement with approximations like EK-FAC while offering better computational scalability and robustness.

**Strengths:**

1. The paper introduces a principled bayesian generalization of classical influence functions, replacing Hessian inversion with covariance estimation over a local posterior, thus making influence computation well defined even for singular deep networks. The method is architecture agnostic, works on any differentiable model. In addition, it scales to models with billions of parameters, avoiding the cubic complexity and instability of Hessian based methods like EK-FAC.
2. The authors rigorously connect BIF to classical IF through Laplace approximation, showing that local BIF recovers damped IFs as a first-order limit and extends them to higher-order geometric effects. And experiments on large models (Pythia-2.8B and Inception-V1) models demonstrate that BIF achieves good data attribution performance.

**Weaknesses:**

1. Although the method avoids explicit Hessian inversion, it still requires running multiple long SGLD chains and forward passes over both training and query sets, making it computationally intensive and potentially slower than classical methods.
2. The theoretical connections to classical IF rely on Laplace approximations and assume locally Gaussian posteriors, which are often invalid for highly non-convex neural loss landscapes. Hence, the higher-order generalization claim might not fully hold in real world networks.

**Questions:**

see weaknesses.

---

> ### Author Response · Authors · 2025-11-24
>
> Thank you for your assessment. We appreciate your recognition of our principled Bayesian generalization and scalability demonstrations.
>
> **W1 (Computational cost)**: We agree that the BIF requires running SGLD chains, which incurs a marginal cost per draw. However, we explicitly analyze this trade-off in Section 3.3 and Table 1\. The critical distinction is that the BIF has no upfront "fitting" phase.
>
> * **EK-FAC**: Requires an expensive pre-computation step involving eigendecompositions that scale cubically ($O(d\_{layer}^3)$) and incurs substantial storage overhead (often hundreds of GBs for large models).
> * **BIF**: Requires zero storage for inverse factors and zero pre-computation time.
>   For the regime of large models (where storing Hessian factors is prohibitive) or fine-grained tasks like per-token influence (where BIF computes the full set of queries in parallel), this trade-off strongly favors the BIF. As demonstrated in our scaling analysis (Fig. 4), this allows the BIF to scale to model sizes where the memory overhead of classical approximations becomes a bottleneck.
>
> **W2 (Laplace approximation validity):** We would like to clarify the scope of our theoretical claim. It is true that the link between the BIF and the classical IF relies on a Laplace approximation, and that this approximation relies on regularity assumptions that do not hold for deep neural networks (unique, non-degenerate minima). However, because the BIF asymptotically (1) recovers the classical IF when these regularity assumptions hold and (2) remains theoretically valid even when they do not, we consider it a natural generalization of the concept. We do not claim that the BIF is the *unique* possible generalization (as other generalizations could surely be constructed). Rather, it is the specific generalization that arises when one treats influence as a property of the full posterior distribution rather than a point estimate. We believe the current manuscript reflects this distinction without overstating the uniqueness of the derivation, but we are happy to clarify this nuance further in the final version if desired.
>
> **Request for clarification:** The concerns you raise (computational cost, theoretical assumptions) appear to be addressed explicitly in our paper (Sec. 3.3, Table 1). Could you clarify if there are specific missing analyses that would move your assessment toward acceptance? We believe the current manuscript establishes the BIF as a robust alternative precisely where classical assumptions fail.

---

### Official Review · Reviewer_AP4E · 2025-10-30

**Soundness:** 3
**Presentation:** 3
**Contribution:** 2
**Rating:** 6
**Confidence:** 2

**Summary:**

The paper introduces the Bayesian Influence Function (BIF), replacing the ill-posed inverse Hessian in classical influence functions with a local posterior covariance estimated via SGLD around the trained weights. On LDS benchmarks, BIF matches strong baselines while avoiding Hessian approximations, and scaling studies show favorable wall clock time compared to EK-FAC on larger models.

**Strengths:**

The paper is well motivated, and the proposed Bayesian Influence Function (BIF) is natural and conceptually sound; the theoretical results clearly establish its relationship to the classical influence function. The scaling demonstrated by BIF is quite favorable.

**Weaknesses:**

1. It is surprising to see sign flips between BIF and EK-FAC, given that both methods are intended to approximate classical influence function. Although the authors provide some explanations, I am not fully satisfied with them. For me, this suggests one of the methods is approximating IF badly. Could the authors add comparisons to IF ground truth? Smaller-scale models are acceptable.
2. To prove practical value, please consider adding at least one downstream evaluation where BIF guides an actionable intervention and improves an objective, especially for the language model settings. The Pythia results are currently limited to qualitative visualizations, which are insufficient to establish the method's usefulness.

**Questions:**

1. How sensitive is BIF to hyperparameters?
2. This method seems to be distributional in nature. Can authors make some connection/comparison with the recent work on distributional data attribution [1]?

[1] Mlodozeniec, Bruno, et al. "Distributional Training Data Attribution." arXiv preprint arXiv:2506.12965 (2025).

---

> ### Author Response · Authors · 2025-11-24
>
> Thank you for your thoughtful and attentive review.
>
> **W1 (Sign flips between BIF and EK-FAC):** We agree that sign flips (e.g., in Figure 4\) are counterintuitive. However, we believe this partly reflects the inherent complexity of defining “influence” in deep neural networks rather than an error in the BIF. As noted in the EK-FAC paper \[1, p. 20\], even for established methods, signs are notoriously difficult to interpret:
>
> > "We also caution the reader that the signs of the influence... tend to be hard to interpret... we have not been able to derive very much insight from whether individual tokens have a positive or negative influence."
>
> Given that visual interpretation is subjective and potentially misleading, we rely on quantitative validation to determine correctness. The ultimate test of sign correctness is retraining: if the BIF were assigning incorrect signs (e.g., classifying a harmful point as helpful), it would result in a negative or poor correlation with ground-truth retraining outcomes. The fact that BIF consistently achieves high Linear Datamodeling Scores (LDS) that match or exceed EK-FAC confirms that its signed estimates are predictive of the actual effect on the model, even where they diverge from the EK-FAC approximation.
>
> **W2 (Downstream evaluation):** We appreciate this suggestion and are conducting additional LDS experiments in a finetuning setting (see top-level comment).
>
> **Q1 (Hyperparameter sensitivity):** Regarding the hyperparameter sensitivity, there is a section in Appendix Figures 9–11. As shown there, the LDS scores remain broadly stable as long as the hyperparameters stay within the same order of magnitude. We therefore conclude that the dependence on hyperparameters is marginal when they are chosen reasonably. We have corrected the reference in the main text (Section 3.1) to point explicitly to Appendix C.3 to ensure this analysis is easily found.
>
> **Q2 (Connection to distributional attribution):** Thank you for this very relevant reference\! There is definitely a connection to the d-TDA paper \[2\]:
>
> Both papers recognize that influence should be viewed probabilistically in terms of the effect data has on the *distribution* of weights. However, they differ in scope and methodology:
>
> 1. **Scope:** d-TDA takes a more agnostic view, defining influence generally as the difference between the unperturbed and perturbed distributions (for arbitrary choices of distribution) using various metrics (e.g., Wasserstein, KL, or Mean shift). The BIF is more specific: it assumes a tempered Gibbs posterior and focuses strictly on measuring the shift in the expectation of an observable. Because d-TDA is defined so broadly, it actually encompasses both the classical IF and the BIF as special cases.
> 2. **Link to classical IF:** Both papers include an asymptotic result linking d-TDA or the BIF to classical influence functions, but these are set up slightly differently and with different assumptions. The d-TDA paper shows that the classical IF approximates the optimal transport map minimizing the KL-divergence between Boltzmann distributions. They use this result (along with a different asymptotic result) to argue why the classical IF might work so well empirically, even in deep neural networks, away from strong regularity assumptions. In contrast, we show that classical IF appears as the leading-order term in the Taylor expansion of the BIF for non-singular models.
> 3. **Methodology:** In the language of d-TDA, the BIF is a “mean-shift” version of d-TDA under a Gibbs measure. This more restrictive definition of the BIF has an advantage: it allows us to use the “covariance trick,” which yields a Hessian-free estimator based on SGMCMC sampling. This differs from the d-TDA paper, which relies methodologically on the existing techniques of unrolled differentiation or Hessian inversion to approximate the distributional shift.
>
> We've expanded Sec. 5 paragraph 2 on “Bayesian influence functions” into a paragraph on “Distribution training data attribution” with commentary on these points.
>
> \[1\] Grosse, Roger, et al. “Studying Large Language Model Generalization with Influence Functions.” arXiv preprint arXiv:2308.03296 (2023).
> \[2\] Mlodozeniec, Bruno, et al. "Distributional Training Data Attribution." arXiv preprint arXiv:2506.12965 (2025).

---

### Official Review · Reviewer_kYHm · 2025-10-31

**Soundness:** 3
**Presentation:** 3
**Contribution:** 3
**Rating:** 4
**Confidence:** 4

**Summary:**

This paper introduces local Bayesian Influence Functions (BIF) for the training data attribution task. BIF estimates data influence through Bayesian covariance sampling using stochastic gradient Markov chain Monte Carlo. The method works for large models, avoids numerical instability, and provides fine-grained, per-token attribution. Experiments on vision and language models show that BIF matches or outperforms some existing methods while being more scalable and flexible.

**Strengths:**

1. The paper introduces the local Bayesian Influence Function with both theoretical grounding and empirical validation.

2. The proposed estimator is architecture-agnostic and scales to billions of parameters.

3. The authors provide thorough implementation and experimental details, which support reproducibility.

**Weaknesses:**

1. The experimental evaluation could be strengthened by including more comparison methods and a broader range of datasets.

2. A robustness analysis or ablation study on key hyperparameters would further enhance the credibility of the results.

3. A brief description of the datasets used would improve the paper’s clarity and completeness.

**Questions:**

Why was Pythia-2.8B chosen over more widely used or newer LLMs?

---

> ### Author Response · Authors · 2025-11-24
>
> Thank you for your review. We appreciate your recognition of our theoretical grounding and and the thoroughness of our implementation details.
>
> **W1 (More datasets/comparison methods):** We're running additional experiments (see top-level comment) to demonstrate generalization beyond CIFAR-10 and the Pile. To respond to the more specific requests:
>
> * **Additional comparison methods (baselines):**  As we detail in the new Appendix A.3, we selected our baselines to ensure coverage across the spectrum of influence approximation strategies: EK-FAC represents state-of-the-art parametric Hessian approximations, TRAK represents "corrected" gradient-based comparisons, and GradSim represents raw gradient comparisons. Since the BIF consistently matches or outperforms these representative approaches on the LDS benchmark (Fig. 3), we believe our evaluation effectively situates the BIF within the current methodological landscape without the need for exhaustive comparison against every variant.
> * **Additional comparison methods (benchmarks):** We focused on the Linear Datamodeling Score (LDS) because it directly measures the correlation between attribution scores and ground-truth retraining outcomes. Unlike heuristic proxy tasks (e.g., mislabel detection or label flipping), LDS quantifies the method’s fundamental ability to predict counterfactual model behavior. Consequently, it has been widely adopted as the standard validation metric in recent literature \[e.g., 1, 2, 3\].
>
> **W2 (Hyperparameter robustness):**  We agree that robustness is critical. This analysis is extensively covered in Appendix C.3 and Figures 9–11, where we demonstrate that the BIF's LDS remains stable across a wide range of SGLD hyperparameters (inverse temperature $\\beta$, localization $\\gamma$, and burn-in steps). We have corrected the reference in the main text (Section 3.1) to point explicitly to Appendix C.3 to ensure this analysis is easily found.
>
> **W3 (Dataset descriptions):**  To ensure the paper is fully self-contained, we have updated Section 4.1 and Section 4.2 to include brief, explicit descriptions of the task specifications for all training and query datasets (e.g., CIFAR-10, The Pile). This information is also available in the reproducibility statement and in the appendices in Table 3\.
>
> **Q1 (Why Pythia-2.8B?):** Unlike many other large models, Pythia offers fully public training data with known ordering (The Pile) and consistent architecture across model sizes (14M to 12B). This transparency is important because IF methods require access to the training data distribution a model was trained on (at least theoretically; practically, we may occasionally be able to get away with slightly different distributions). We chose Pythia-2.8B out of a practical trade-off between scale and compute/iteration speed.
>
> **Request for clarification:** We are actively working to expand our empirical validation to additional models and datasets (as described in the top-level comment). Would demonstrating strong LDS performance on a finetuning tasks (e.g., GPT-2 on WikiText-2 or larger Pythia models) be sufficient to address your concerns, or are there other specific experiments you consider essential for acceptance?
>
> \[1\] Bae et al., “Training Data Attribution via Approximate Unrolled Differentiation” arXiv preprint arXiv:2405.12186
> \[2\] Park et al., “TRAK: Attributing Model Behavior at Scale” arXiv preprint arXiv:2303.14186
> \[3\] Wang et al., “Better Training Data Attribution via Better Inverse Hessian-Vector Products” arXiv preprint arXiv:2507.14740

---

### Official Review · Reviewer_hnax · 2025-11-04

**Soundness:** 4
**Presentation:** 4
**Contribution:** 3
**Rating:** 8
**Confidence:** 4

**Summary:**

The paper introduces the Local Bayesian Influence Function (BIF), a novel approach to Training Data Attribution (TDA) for deep neural networks. It addresses the limitations of classical influence functions, which rely on Hessian inversion and struggle with the non-invertible Hessians and high-dimensional parameter spaces of modern deep learning models. The proposed BIF replaces Hessian inversion with covariance estimation over the local posterior, leveraging stochastic-gradient MCMC sampling for efficient computation. The method is architecture-agnostic and scales to models with billions of parameters. Empirical results demonstrate that BIF achieves state-of-the-art performance in predicting retraining experiments and offers computational advantages over classical methods like EK-FAC, particularly for fine-grained and per-token attribution tasks.

**Strengths:**

- The paper provides a principled extension of classical influence functions to the Bayesian setting, addressing the challenges posed by non-invertible Hessians in deep neural networks. The proposed BIF method also scales efficiently to models with billions of parameters, making it suitable for modern large-scale deep learning architectures.

- The BIF enables per-token influence computation, which is particularly useful for language models and provides insights into semantic relationships between tokens. Moreover, unlike classical methods, BIF can be applied to any differentiable architecture, including attention-based models, making it more versatile.

- The paper provides a thorough comparison with state-of-the-art methods like EK-FAC, highlighting the advantages and trade-offs of the BIF approach.

**Weaknesses:**

-  While BIF avoids the high up-front fitting cost of EK-FAC, its computational cost scales with the number of posterior draws, which can be expensive for large datasets or models.

- While the BIF provides interpretable results in many scenarios, there are cases, particularly in language modeling, where the most influential samples are not immediately intuitive.

- The authors are requested to provide some initial experiments on more modern deep learning models and slightly larger models.

- The paper primarily focuses on comparing BIF with EK-FAC, leaving other TDA methods like TRAK and GradSim less explored in depth.

**Questions:**

Please see the Weaknesses.

Overall, the paper makes a significant contribution to the field of AI interpretability and training data attribution by introducing a scalable, theoretically grounded method that addresses key limitations of classical influence functions. The empirical results are promising, and the method's ability to handle large-scale models and provide fine-grained attribution is a notable advancement.

---

> ### Author Response · Authors · 2025-11-24
>
> Thank you for your thoughtful and positive review. We are pleased that you found the BIF framework compelling and appreciated the demonstrations of its scalability.
>
> **W1 (Computational cost):** We acknowledge that BIF’s cost scales with the number of posterior draws, which is a limitation when evaluating large datasets. However, we emphasize that this does not pose a specific challenge for *large models*. On the contrary, the trade-offs (specifically the absence of an expensive upfront “fitting” phase) favor the BIF in two regimes: (1) large models where storing/inverting Hessian approximations is prohibitive, and (2) fine-grained analysis, such as per-token influence, where BIF’s ability to batch queries offers significant parallelism. This is detailed in Sec. 3.3 and Table 1\.
>
> **W2 (Interpretability in language modeling):** While we agree that BIF results are not always immediately human-interpretable, we believe this reflects a fundamental property of influence functions rather than a limitation specific to the BIF. As noted by Grosse et al. \[1, p. 5\], leading techniques like EK-FAC often highlight training examples that are not semantically intuitive (and instead seem to reflect low-level statistical patterns).
>
> **W3 (Larger/more modern models):** We are running additional experiments. See our top-level comment for more.
>
> **W4 (Broader TDA method comparison):** We have added Appendix A.3 to explicitly contextualize the BIF against TRAK and GradSim. As detailed there, while these methods are valuable, they either require specific architectural approximations (e.g., random projections for TRAK) or lack the geometric correction of the Hessian (GradSim). Our benchmarks (Fig. 3\) demonstrate that BIF consistently matches or outperforms these baselines on the LDS metric while remaining fully architecture-agnostic.
>
> We hope these revisions and clarifications address your concerns. We are happy to engage further if you have additional questions during the discussion period.
>
> \[1\] Grosse, Roger, et al. “Studying Large Language Model Generalization with Influence Functions.” arXiv preprint arXiv:2308.03296 (2023).

---

### Author Response · Authors · 2025-11-24

We thank all reviewers for their very constructive feedback. We are encouraged by the consensus that our work provides a principled generalization of influence functions (hnax, UhDc, AP4E, kYHm) that scales effectively to billion-parameter models (hnax, UhDc, AP4E, kYHm) while remaining architecture-agnostic (hnax, UhDc, kYHm).

We address three common themes below:

**1\. Computational trade-offs and scaling (hnax W1, UhDc W1)**

*Reviewers noted that the iterative nature of SGLD chains incurs a high marginal cost per attribution, potentially making the BIF computationally intensive compared to classical methods that amortize cost over many queries.*

We acknowledge this weakness and explicitly analyze these trade-offs in Section 3.3 and Table 1\. The key insight is that the BIF and classical IF approximations (like EK-FAC)  have complementary cost structures:

* **EK-FAC:** Incurs a high up-front fitting cost (scaling cubically, $O(d\_{layer}^3)$, per layer) and significant memory overhead for storing inverse factors, but offers a lower marginal cost per query once fitted.
* **BIF**: Has zero up-front fitting or storage costs, but scales linearly with the number of posterior draws required for convergence.
  Our benchmarking (Fig. 4\) demonstrates that the BIF becomes faster and more memory-efficient than EK-FAC at scale, particularly for fine-grained attribution tasks like per-token influence, where the BIF allows the full influence matrix to be computed in parallel rather than sequentially.

To more explicitly compare the BIF against other IF approximations besides just EK-FAC, we have added an Appendix A.3 and Table 2, which details further comparisons (especially against TRAK and GradSim).

**2\. Hyperparameter robustness (kYHm W2, AP4E Q1)**

*Reviewers questioned the sensitivity of the method to SGLD hyperparameters (e.g., inverse temperature, localization strength) and requested evidence that the method is stable without extensive tuning.*

We agree that robustness to hyperparameters is important. This analysis is comprehensively covered in Appendix C.3 and Figures 9–11, which show that the BIF's Linear Datamodeling Score (LDS) remains stable across wide ranges of hyperparameters (inverse temperature $\\beta$, localization $\\gamma$, burn-in) provided the SGLD chains converge.

We have updated the main text (Section 3.1) to correct a reference error that made these hyperparameter ablations difficult to find.

**3\. Expanded empirical validation & downstream application (hnax W3, kYHm W1, AP4E W2)**

*Reviewers requested validation on a broader range of models (esp. larger and more modern) and datasets to ensure generalization beyond the initial scope, as well as demonstrations of utility in practical downstream settings.*

**Regarding downstream utility:** Our LDS retraining experiments (Fig. 3, Sec. 4.2) already demonstrate this directly since they measure BIF's ability to predict real outcomes from data interventions across hundreds of retraining runs.

**Regarding broader empirical validation:** We are conducting LDS experiments in finetuning settings on language models to demonstrate generalization beyond our initial scope. We will include these in the camera-ready version.

We emphasize, however, that our core contribution of establishing the BIF as a theoretically sound, scalable Hessian-free alternative to state-of-the-art classical approximations is validated by our LDS benchmarking and stands independently of these specific downstream tasks.

Please see our individual responses below for point-by-point replies to each reviewer's specific questions.

**Specific Changes:**

The following changes were made following feedback from reviewers:

* **Improve visibility of hyperparameter ablations:** We replaced an incorrect reference to B.1 with a reference to C.3 in Sec. 3.3.
* **Improve visibility of dataset details:** We added explicit references for the datasets we used to Sec. 4.1 in the last two paragraphs and to Sec. 4.2 in paragraph 2\.
* **Add link to distributional TDA:** We expanded section 5 paragraph 2 on “Bayesian Influence Functions” with discussion on distributional TDA and renamed this paragraph to “Distributional Training Data Attribution.”
* **Add discussion of other IF approximations (TRAK and GradSim):** We added App. A.3 with additional explanation of how the BIF compares against other IF techniques. We also added forward references to this section from the appendix outline, Sec. 3.3, and from Table 1\.

We also took the opportunity to fix a few typos/formatting issues.

---

### Meta-Review · Area_Chair_kAaa · 2026-01-07

**Summary:**

This paper introduces Bayesian Influence Functions (BIF), a Hessian-free and architecture agnostic framework for training data attribution that scales to modern deep networks via stochastic-gradient MCMC. Reviewers agree that the approach is principled, timely, and empirically effective on retraining-based benchmarks. While concerns were raised regarding computational trade-offs, theoretical assumptions, and the breadth of empirical validation, the rebuttal and additional experiments addressed most practical issues and clarified the method’s scope. Overall, the contribution is considered solid for large-scale data attribution and interpretability.

**Reviewer Concerns:**

Addressed Concerns:

**Computational trade-offs and scaling (Reviewer hnax, UhDc).** The authors clarified that, unlike EK-FAC and related methods, BIF has no expensive upfront fitting or storage costs, and demonstrated through scaling experiments that it becomes more efficient and memory-
friendly at large model scales and for fine-grained attribution tasks.

**Hyperparameter robustness (Reviewers kYHm, AP4E).** Concerns about sensitivity to SGLD hyperparameters were addressed through extensive ablation studies showing stable attribution performance across wide parameter ranges, provided the chains converge.

**Comparison to other TDA methods (Reviewers hnax, kYHm).** The authors address this concern by adding explicit comparisons with TRAK and GradSim, and the results demonstrate that BIF consistently matches or outperforms representative alternatives on
retraining-based metrics while remaining fully architecture-agnostic.

**Expanded empirical validation and downstream application (Reviewers hnax, kYHm, AP4E).** The authors argue that the LDS retraining experiments already demonstrate downstream utility and indicate that broader empirical validation will be included in the camera-ready version.

Outstanding Concerns:

**Laplace approximation validity (Reviewer UhDc).** The authors did not fully address concerns about whether the higher-order generalization claims hold in real-world networks where Laplace assumptions may fail. While they clarify that BIF is not intended as a unique generalization, the paper still lacks a clearer discussion of the practical validity and limitations of the Laplace approximation.

**Reviewer Scores:**

**Reviewer hnax:** Score likely remains unchanged (8), as concerns regarding computational cost, broader comparisons and experiments, and interpretability in language modeling have been fully addressed.

**Reviewer kYHm:** Score is likely to improve (currently 4), as concerns regarding broader experimental coverage and hyperparameter robustness have been addressed through additional experimental results.

**Reviewer AP4E:** Score likely remains unchanged (6), as the authors provide a detailed discussion of sign flips between BIF and EK-FAC and address concerns about downstream evaluation through additional experiments.

**Reviewer UhDc:** Score likely remains unchanged (4), as concerns regarding the validity of the Laplace approximation remain insufficiently addressed.

---

### Decision · Program_Chairs · 2026-01-26

Accept (Poster)